# CRISPECTOR provides accurate estimation of genome editing translocation and off-target activity from comparative NGS data

Ido Amit [1,5], Ortal Iancu[2,5], Alona Levy-Jurgenson[3], Gavin Kurgan [4], Matthew S. McNeill[4], Garrett R. Rettig [4], Daniel Allen[2], Dor Breier[2], Nimrod Ben Haim[2], Yu Wang[4], Leon Anavy[1], Ayal Hendel [2✉] & Zohar Yakhini [1,3✉]

Controlling off-target editing activity is one of the central challenges in making CRISPR technology accurate and applicable in medical practice. Current algorithms for analyzing off-target activity do not provide statistical quantification, are not sufficiently sensitive in separating signal from noise in experiments with low editing rates, and do not address the detection of translocations. Here we present CRISPECTOR, a software tool that supports the detection and quantification of on- and off-target genome-editing activity from NGS data using paired treatment/control CRISPR experiments. In particular, CRISPECTOR facilitates the statistical analysis of NGS data from multiplex-PCR comparative experiments to detect and quantify adverse translocation events. We validate the observed results and show independent evidence of the occurrence of translocations in human cell lines, after genome editing. Our methodology is based on a statistical model comparison approach leading to better false-negative rates in sites with weak yet significant off-target activity.

[1] Arazi School of Computer Science, Interdisciplinary Center, Herzliya, Israel. [2] The Institute for Advanced Materials and Nanotechnology, The Mina and Everard Goodman Faculty of Life Sciences, Bar-Ilan University, Ramat-Gan, Israel. [3] Department of Computer Science, Technion—Israel Institute of Technology, Haifa, Israel. [4] Integrated DNA Technologies Inc., Coralville, IA, USA. [5] These authors contributed equally: Ido Amit, Ortal Iancu. ✉email: ayal.hendel@biu.ac.il; zohar.yakhini@idc.ac.il

Clustered Regularly Interspaced Short Palindromic Repeats (CRISPR) genome editing, which utilizes guide RNA (gRNA)-directed Cas nucleases to induce double-strand breaks (DSBs) in the treated genome, has shown promising preliminary results as an approach for definitively curing a variety of genetic disorders[1]. It is important to remember that CRISPR-Cas endonucleases did not naturally evolve to function as a highly specific gene-editing mechanism, certainly not in the context of mammalian genomes. Using these bacterial nucleases in mammalian, plant, and other types of cells often entails off-target activity, leading to unintended DNA breaks at other sites in the genome with only partial complementarity to the gRNA sequence[2]. To bring CRISPR technology, or any other engineered nuclease, to safe and broader use in the clinic, it must be highly active at the on-target site and have minimal off-target editing adverse effects. These system properties are also highly desirable for the use of genome editing in crop engineering and other applications[3–6]. The Cas9 endonuclease can create DSBs at undesired off-target locations, even in the presence of mismatches. This off-target activity creates potential major pitfalls for CRISPR-based therapies due to unwanted genome instability events such as resulting point mutations and genomic structural variations. Therefore, selecting systems with improved accuracy, precision, and specificity, in the context of a given application is of major significance. Several methodologies have been developed to detect off-target activity in an unbiased manner[7]. One such approach is Genome-wide, Unbiased Identification of DSBs Enabled by Sequencing (GUIDE-seq)[8]. GUIDE-seq relies on the introduction of short tag DNA sequence that is transfected into the cells and integrated at the Cas9-induced DSB via non-homologous end joining (NHEJ). The subsequent sequencing of tag-adjacent regions identifies potential off-target sites. Other approaches include Circularization *In Vitro* Reporting of Cleavage Effects by Sequencing (CIRCLE-seq)[9], Selective Enrichment and Identification of Adapter-Tagged DNA Ends by Sequencing (SITE-seq)[10], and Discovery of *In Situ* Cas Off-Targets and Verification by Sequencing (DISCOVER-Seq)[11], among others. Potential off-target sites, detected by the unbiased genome-wide approaches mentioned above, can then be validated and quantified by targeted amplicon sequencing with primers designed for the reported genomic loci (e.g., rhAmpSeq[12,13]).

One group of possible consequences of off-target editing, which can be particularly devastating even when occurring at low frequencies, are structural variations that lead to translocations and other adverse fusion events. Deleterious genomic structural variation events can lead to the onset of several human disease conditions including many types of cancer, infertility, and other acquired genomic disorders[14,15]. Chromosomal translocations can arise when on-target/off-target or off-target/off-target cut-sites fuse as a result of DSBs at both loci[16]. Therefore, translocations and other adverse structural effects of editing require thorough investigation to improve our understanding of their prevalence, their characteristics, and the conditions promoting their formation. Previously published methods for de novo translocation detection, Anchored Multiplex-PCR sequencing (AMP-Seq)[8,17], High-Throughput, Genome-wide, Translocation Sequencing (HTGTS)[18], and Uni-Directional Targeted Sequencing (UDiTaS)[19], identify translocations involving selected DSB sites (on-target or off-target) inflicted by genome editing using site-directed nucleases. Since translocations can involve various combinations of off-target loci or occur between the off-target loci and spontaneous breaks, the ability to detect translocations on the entire spectrum of DSBs is essential.

As in other fields of genomics and genetics where efficient and intuitive visualization and analysis tools are central to practice and progress[20–23], tools and methods for the interpretation and visualization of genome editing are being developed. More than 10 bioinformatics analysis tools, developed for the purpose of estimating genome-editing activity rates from experimental data, have been previously published[24–29]. In the process of generating NGS sequencing data, a number of errors are introduced and then represented in the raw sequencing data. These errors come from various sources, including the sequencing platform, polymerase errors, and library preparation buffers (e.g., base oxidation)[30]. These events can occur near or at the site of the CRISPR-induced DSB, limiting the accuracy of bioinformatics analysis tools aimed to quantify off-target events with low activity rates, as it is difficult to distinguish NGS errors from actual edit events. To support the detection of lower-level activity a treatment (Tx) vs mock (M) approach is often taken[12]. CRISPResso1/2[31,32] are currently the most popular tools used, in practice, in this domain. The approach taken by CRISPResso to address Tx vs M experiments is based on subtracting the inferred rates at each of the conditions separately. The combined Tx vs M insertion or deletion (indel) rate at a given site is used, not stratified into different types. This can be problematic since a greater incidence of indel events of a certain type that is unique to Tx may be masked by other types. Another popular tool is ampliCan[33], which is the first tool to purportedly directly address the Tx vs M design. ampliCan's published description, however, has only demonstrated its performance, for real measurement data, on singleplex experiments, with no performance indications as related to detecting off-target activity. ampliCan cleans noisy observations and then also applies subtraction to infer activity rates. ampliCan separately addresses different modification types, thereby avoiding the masking issue described above. In addition, as these tools perform simple background subtraction they provide no statistical estimate of the inferred rates or the obtained differences. The use of confidence intervals is central to other domains in genetics and genomics. It is also important to note that most of the existing methods are not designed to smoothly process multiplex experiments, in which multiple potential targets can be analyzed. Finally, and maybe most importantly in the context of this work, the detection of genome-editing-associated translocations is not supported with any previously published genome-editing quantification and data-analysis tool.

The tool and the underlying methods we present herein, called CRISPECTOR, constitute a convenient platform for analyzing Tx vs M multiplex-PCR data that combines superior performance in measuring off-target activity with a built-in module for translocation detection.

CRISPECTOR represents progress over the state-of-the-art in four principal aspects: (i) increased accuracy based on Tx vs M experiments enabled by the use of a statistical model comparison approach (ii) supported detection of alternative cut-sites in off-target loci (iii) analysis of multiplex-PCR and NGS data to detect adverse structural variations and translocation events occurring in an editing experiment, and (iv) reporting statistical confidence intervals for inferred off-target activity rates.

## Results

**CRISPECTOR—a tool for measuring and quantifying off-target genome editing activity.** CRISPECTOR is a software package that accepts FASTQ files resulting from running Tx vs M experiments followed by multiplex-PCR and NGS. A configuration table with details of the experimental parameters is also part of the input (Supplementary Fig. 1). The tool analyzes the NGS input and applies statistical modeling to determine and quantify NHEJ-mediated indel (henceforth often abbreviated as indel) edit activity at every interrogated locus, as well as adverse translocation activity in all relevant pairs of loci. Statistical confidence indicators are computed for the reported results, enabling us to provide a better estimation of the potential sites in which off-target activity can be expected. CRISPECTOR is designed to

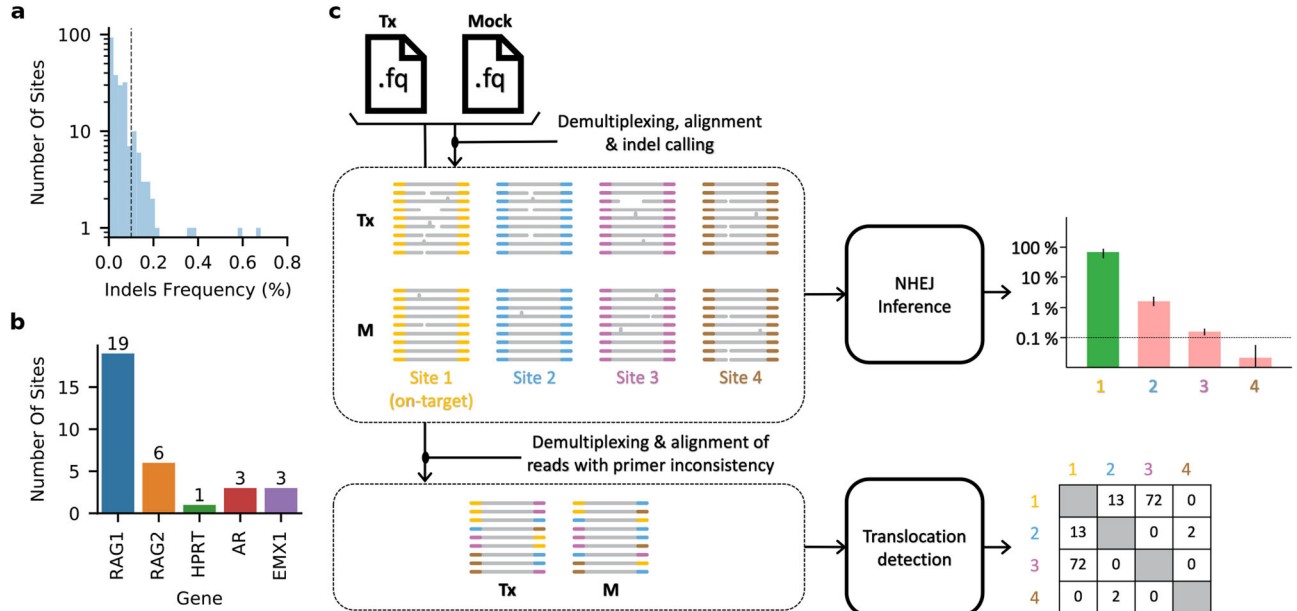

**Fig. 1 Quantifying off-target activity by CRISPECTOR. a, b** Hypothetical indels at the expected cut-site in the M experiments. **a** Out of 226 off-target sites that we have examined in our experimental data, we have found 31 sites to have indel frequency higher than 0.1% (fraction = 0.001) in the M experiments. M indel frequencies at the cut-site were measured by a direct calculation, not using a comparative set-up. **b** These 31 noisy off-target sites come from different gRNAs examined, as depicted. (**c**) CRISPECTOR workflow. CRISPECTOR assigns each read in the Tx and M FASTQ files to a specific locus of interest or a putative translocation. Then, a Bayesian inference classifier accurately estimates the indel editing activity, and a hypergeometric test is performed to detect translocation reads.

be highly sensitive, detecting events at rates as low as 0.1%. Such events occur quite often, as shown in Fig. 1a, b. The workflow is depicted in Fig. 1c. CRISPECTOR generates an HTML-based report to support user interpretation and further analysis of the outcomes. Supplementary Note 1 includes an installation and user manual together with detailed examples of the input files and the output report. CRISPECTOR is available for download in http://bioconda.github.io/recipes/crispector/README.html (see "code availability section").

To assess CRISPECTOR's accuracy we applied it to multiplex-PCR NGS data produced from five different gRNAs, corresponding to five on-target genomic loci, and covering 226 off-target sites, under different experimental conditions. This yields a total of 1161 instances (see the experimental data section in "Methods"). The experiments use a rhAmpSeq assay (IDT, Coralville, IA) for the PCR in a Tx vs M design (more details in "Methods"). We present results obtained from evaluating indel and translocation activity in these experiments.

**CRISPECTOR accurately estimates indel activity levels at off-target sites.** CRISPECTOR editing activity estimation is based on the ability to model the experimental background noise. The core classification algorithm learns the specific pattern of background noise for each experiment from the sequencing data and utilizes a Bayesian approach to call and quantify editing events. The core classification is blind to the source of the noise, whether it comes from high NGS error rates (Fig. 1a, b), false site assignments, or ambiguous alignments.

We test our performance on the most challenging off-target scenarios, where high error rates occur at sites with low editing activity and show that CRISPECTOR can often recover the true validated editing activity even when error and editing rates are near identical. Validation is obtained by human examination of the actual reads as well as by two lines of statistical evidence as explained in the "Methods" and in the "Discussion" section. Supplementary Fig. 2 depicts indel editing activity results for a

RAG2 experiment in HEK293 with stable expression of Cas9 (abbreviated HEK293-Cas9) including multiple off-target sites with low editing rates detectable by CRISPECTOR. We observed that off-target indel activity levels in candidate sites with less homology to the on-target gRNA were generally lower (Supplementary Fig. 3). An additional example of analysis results, for RAG1 in the same conditions, is provided in Supplementary Fig. 4.

Process-related (NGS, PCR, etc.) error rates can lead to false-negative inferences by existing tools; higher rates of M indels can hide real editing activity. An example of this can be seen at off-target site 51 in the RAG1 experiment (Fig. 2a) where M indels surpass those of real editing activity. CRISPECTOR classified as edited 32 reads with deletions of variable lengths, and it estimated an editing activity of 0.113% overcoming the strong background noise. The reads that support this were individually validated by human inspection and are fully reported in Supplementary Note 2. An elevated level of 41 reads with deletions of 1 bp length at the expected cut-site in M, caused CRISPResso2[32] and ampliCan[33] to estimate a much lower signal, 0.068% and 0.07%, respectively. Similarly, higher rates of noise around the cut-site can lead to false activity calling by existing tools. For example, off-target site 37 in a different RAG1 experiment (Supplementary Fig. 5). At this locus, CRISPECTOR inferred a 0% editing activity while CRISPResso2 and ampliCan estimated 0.136% and 0.361%, respectively. This apparent miscalling by existing tools probably results from deletions with a length of 5 bp at the cut-site, 29 in Tx compared to 19 in M. Such differences are not sufficiently extreme, statistically, and these reads are correctly classified as noise by our classifier.

Accurate editing activity estimation also depends on the detection of alternative cut-sites. Flawed identification of the off-target gRNA binding configuration, due to an ambiguous alignment or false interpretation of GUIDE-seq or other screening methods, can lead to a mis-inferred PAM position. Moreover, optimal read alignments, even if slightly better justified from a

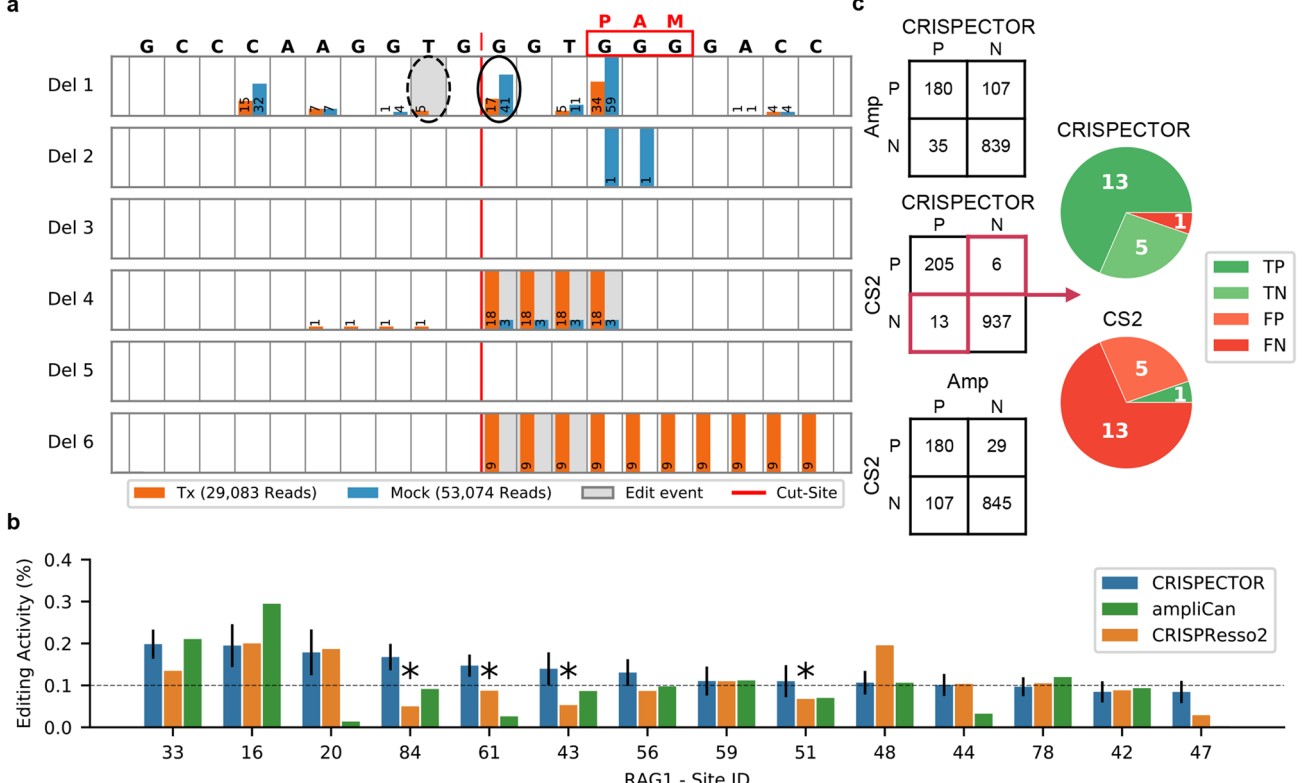

**Fig. 2 Off-target activity levels estimates by CRISPECTOR. a** Site 51 of *RAG1*, XT 2p in HEK293-Cas9. High rate of deletions of length 1 at the expected cut-site masks the real editing signal (Tx=17, M = 41, solid circle) if only subtraction is used. Also, the PAM site is GGG. Thus, we can find an alternative cut-site 1-bp to the right of the expected cut-site. CRISPECTOR points to five reads with deletions of length 1 on this alternative cut-site (dashed circle). Editing activity estimation—CRISPECTOR = 0.113%, CRISPResso2 = 0.068% and ampliCan = 0.070%. Note that reference is depicted 3'→5' since the gRNA cuts the negative strand in this locus. The notation Del **k** stands for deletions of length k. **b** Editing activity by CRISPECTOR, ampliCan and CRISPResso2 for *RAG1*, XT 2p, HEK293-Cas9 sample. For many instances with low editing activity levels, there is a disagreement between tools with respect to the active (≥0.1%) or inactive (<0.1%) classification. In this experiment, CRISPECTOR reported four sites as ≥0.1% (84, 61, 43, and 51, denoted with asterisks), where both ampliCan and CRISPResso2 determined the sites as ≤0.1%. Error bars represent confidence intervals of CRISPECTOR results. This activity was validated by human examination of the individual actual reads (see "Discussion" and Supplementary Note 2). **c** Tool comparison for the active off-target site classification. Off-target site classification results for CRISPECTOR, CRISPResso2 (CS2), and ampliCan (Amp). In the left column squares, the number of instances with agreement and disagreement in the classification task are depicted. Each square refers to a comparison of two tools. P (Positive) and N (Negative) stand for active and inactive sites, respectively. The two pie charts depict CRISPECTOR and CRISPResso2 classification results on the 19 validated sites. TP, TN, FP, and FN denote true-positive, true-negative, false-positive, and false-negative, respectively. True state determined by the actual reads.

**Table 1 Active site classification results comparison between CRISPECTOR to CRISPResso2 and ampliCan.**

|  | RAG1 | RAG2 | AR | HPRT | EMX1 | Total |
|---|---|---|---|---|---|---|
| *CRISPResso2* | | | | | | |
| Agreement | 490 | 298 | 153 | 104 | 97 | 1137 |
| Disagreement | 12 | 1 | 2 | 1 | 3 | 19 |
| *ampliCan* | | | | | | |
| Agreement | 416 | 269 | 142 | 101 | 91 | 1019 |
| Disagreement | 86 | 30 | 13 | 4 | 9 | 142 |

Agreement rows represent agreement on the site/instance classification output, active (>0.1%) vs inactive (<0.1%), between CRISPECTOR and other tools, as defined in "Results". The disagreement row for CRISPResso2 was verified by examination of the actual reads. Validation results in Supplementary Note 2.

biochemical perspective, can place real edit events away from the expected cut-site (Supplementary Note 3). Finally, real editing activity can occur away from the expected cut-site due to the existence of an alternative PAM sequence or less frequent non-canonical DSB mechanisms (see Supplementary Fig. 6 and Supplementary Note 4). CRISPECTOR enables the detection of

alternative cut-sites by incorporating different prior probabilities for each position in the reference sequence, as described in the "Methods" section.

**Comparison with other tools**. We used the five different gRNAs, corresponding to five on-target genomic loci, investigated in this study, to compare CRISPECTOR to other tools, in the context of NHEJ-mediated indel activity. Our data covers 226 off-target sites, under different experimental conditions (human cell types and experimental systems), with a total of 1,161 instances. We analyzed the editing activity estimation concordance between CRISPECTOR, CRISPResso2, and ampliCan. We found that CRISPECTOR and CRISPResso2 produce similar results, with a relative difference median of 13.1%, while ampliCan estimation substantially differs from both CRISPECTOR and CRISPResso2, with a relative difference median of 75.6% and 76.9%, respectively (Supplementary Fig. 7).

We further assessed the editing rate accuracy on the challenging task of active off-target site classification. Editing rates are often used by the community[2] to classify an off-target site as either 0.1% or < 0.1% (within possible experimental noise).

**Table 2 CRISPResso2 and CRISPECTOR errors per gRNA.**

|  | RAG1 | RAG2 | AR | HPRT | EMX1 | Total |
|---|---|---|---|---|---|---|
| *CRISPResso2* | | | | | | |
| FN | 7 | 0 | 2 | 1 | 3 | 13 |
| FP | 3 | 1 | 0 | 0 | 1 | 5 |
| *CRISPECTOR* | | | | | | |
| FN | 1 | 0 | 0 | 0 | 0 | 1 |
| FP | 0 | 0 | 0 | 0 | 0 | 0 |

False-positive (FP) and false-negative (FN) with respect to the classification into active (>0.1%) vs inactive (<0.1%) off-target site/instance. See Supplementary Note 2 for full validation report.

Assessment of such classification requires the use of a gold standard dataset of validated editing rates. In this analysis, we define the validated percent indels as the value determined through a detailed human investigation of the individual raw alignment results. In some instances, differences between the tools are significant (Fig. 2b). We consider a classification call to be correct when the evaluated editing activity for an off-target site is at least 0.1% and the same is true for the human validated call, and vice versa. Since human validation of all 1161 instances is infeasible, we constructed a manually curated set of validated calls consisting of the 19 instances/sites with a disagreement between CRISPECTOR and CRISPResso2 (Fig. 2c, Table 1, and Supplementary Note 2). 18 of these 19 instances were determined, by human expert investigation of the reads, to be CRISPResso2 errors—5 false-positives and 13 false-negatives (Fig. 2c, Table 2, and Supplementary Note 2). CRISPECTOR had one false-negative error, as a result of sequencing background noise. Within the validated 19 instances, 13 instances also represent a disagreement between CRISPECTOR and ampliCan. All 13 instances were determined to be ampliCan errors (2 false-positives and 11 false-negatives). We do not perform the same procedure with ampliCan since the total classification disagreement between CRISPECTOR and ampliCan is 142 instances (Fig. 2c, Table 1; CRISPResso2 and ampliCan have 136 classification disagreements with 127 instances amongst these 142) - too many for an exhaustive examination. Running parameters are described in Supplementary Note 5. Concordance of inferred rates, with respect to the reaction efficiency, as described in the "Methods", also supports (binomial $p$-value < $1.7 \times 10^{-3}$, see Supplementary Note 6) our validation conclusions.

**CRISPECTOR detects evidence of translocations resulting from CRISPR editing activity**. As indicated above, one of the features of CRISPECTOR is its ability to detect translocations resulting from CRISPR editing by analyzing NGS data produced by a multiplex PCR using locus-specific primers (such as rhAmpSeq[34]) (Supplementary Fig. 8). Using this mechanism for target enrichment, four types of translocation events can occur for every pair of potential partner loci (Fig. 3a). For each off-target site, a pair of primers were designed to span the potential cut-site. CRISPECTOR analyzes mixed pairs of primer sequences that are detected on common reads in the NGS data. These reads represent putative fusion amplicons. Once putative fusion amplicon configurations are identified in at least one read, they are scored based on a hypergeometric statistical model (see "Methods") and reported to the user, ordered according to the inferred statistical significance (Supplementary Fig. 9). CRISPECTOR reports all translocations with significant (FDR corrected) $p$-values (≤0.05, by default) as a heatmap plot (Fig. 3b).

In our experiments, significant translocations were found both in experiments editing RAG1 and RAG2 loci in HEK293-Cas9 cells[12]. We found evidence for translocations in 20 and 19 unique pairs of sites, for RAG1 and RAG2, respectively. The most significant corrected $p$-values are $4.9 \times 10^{-22}$ for on-target site 1 with off-target site 7 in RAG1, and $1.53 \times 10^{-53}$ for off-target site 1 with off-target site 5 in RAG2 (Supplementary Fig. 10). We also found evidence of translocations in 9 pairs of sites for the EMX1 experiment in HEK293-Cas9 cells. The most significant corrected $p$-value was $1.8 \times 10^{-162}$ for on-target site 1 with off-target site 2. We noted that translocation events in RAG1 and RAG2 were found in experiments with high coverage. RAG1 and RAG2 experiments have a depth of 3.7 M and 4.2 M reads (with an average of 43K and 79K per site), and the numbers of detected translocation reads are 497 (0.013%) and 599 (0.014%), respectively. However, for EMX1, the percentage of detected translocation reads is 0.35%, 2137 detected translocation reads out of 600K reads (61K per site). This is probably due to the high editing rate of EMX1 off-target sites 2 (68.8%) and 3 (57.6%).

These results were experimentally confirmed in an independent measurement using droplet digital PCR (ddPCR, see Fig. 3c and "Methods") using primers designed to amplify individual potential translocation events. The translocation frequencies determined via ddPCR are consistent with the frequencies found by CRISPECTOR on NGS data, with the most prevalent events being RAG2_1/RAG2_5 for RAG2; RAG1_7/RAG1_1 for RAG1; and EMX1_2/EMX1_ 1 for EMX1. When multiple genomic sites are simultaneously broken, the cell can join the broken ends in a number of different combinations, as long as the basic biological principle of establishing a phosphodiester bond between a 3′ carbon atom of one deoxyribose and the 5′ carbon atom of another deoxyribose prevails (Fig. 3a). We noted that the most frequent RAG2 single observation pertains to a fusion formed between two off-target sites, namely RAG2_1 and RAG2_5. Both loci reside on the q arm of chromosome 11. Amongst the potential events that can be formed due to these two CRISPR DSBs, is the centromere-free product that shows the strongest signal (Supplementary Fig. 11, and Supplementary Table 2) as validated by the PCR experiment. Furthermore, we noted, interestingly, that off-target site RAG2_1 gives rise to more translocations than the RAG2 on-target site (Fig. 3b). Finally, we noted that all four possible translocation configurations that can theoretically arise from aberrant resection of simultaneous DSBs in two loci (Fig. 3a) occurred in our experimental data. This last statement pertains to the type of events and not to the actual occurrence of all four types per pair of loci. The ability to detect all four types at any potential junction depends on the NGS library preparation (see "Discussion").

To further validate and characterize the translocation detection process, serial dilutions of a synthetic fusion translocation construct (RAG2_10/RAG2_7) were spiked-in to achieve 0.016%, 0.16%, 1.6%, and 16% fractions of the total RAG2, XT 2-part gRNA (XT 2p), HEK293-Cas9 genomic DNA. The synthetic fusion construct represents an artificial translocation event between RAG2 off-target sites 7 and 10, in the form of P7adaptor-5′-(RAG2_10)-(RAG2_7)-3′-P5adaptor. These sites were not identified by CRISPECTOR as possible translocation partners in the original data. The samples were analyzed by NGS following multiplex PCR (rhAmpSeq, as in the rest of the experiments described here) and by ddPCR quantification. As shown in Fig. 3d and in Supplementary Fig. 12, we observe an increase in the detected signal for the synthetic construct while a naturally occurring translocation yields a signal with low variance (CV = 0.2). CRISPECTOR detected the RAG2_10/RAG2_7 translocation down to 0.16%, with significant (FDR corrected) $p$-values (≤0.05). The translocation frequency was also measured using ddPCR for the 0.016% spiked-in sample, yielding 0.0169%. There was no detection of the synthetic fusion construct in the control experiments (un-spiked RAG2, XT 2p, HEK293-Cas9

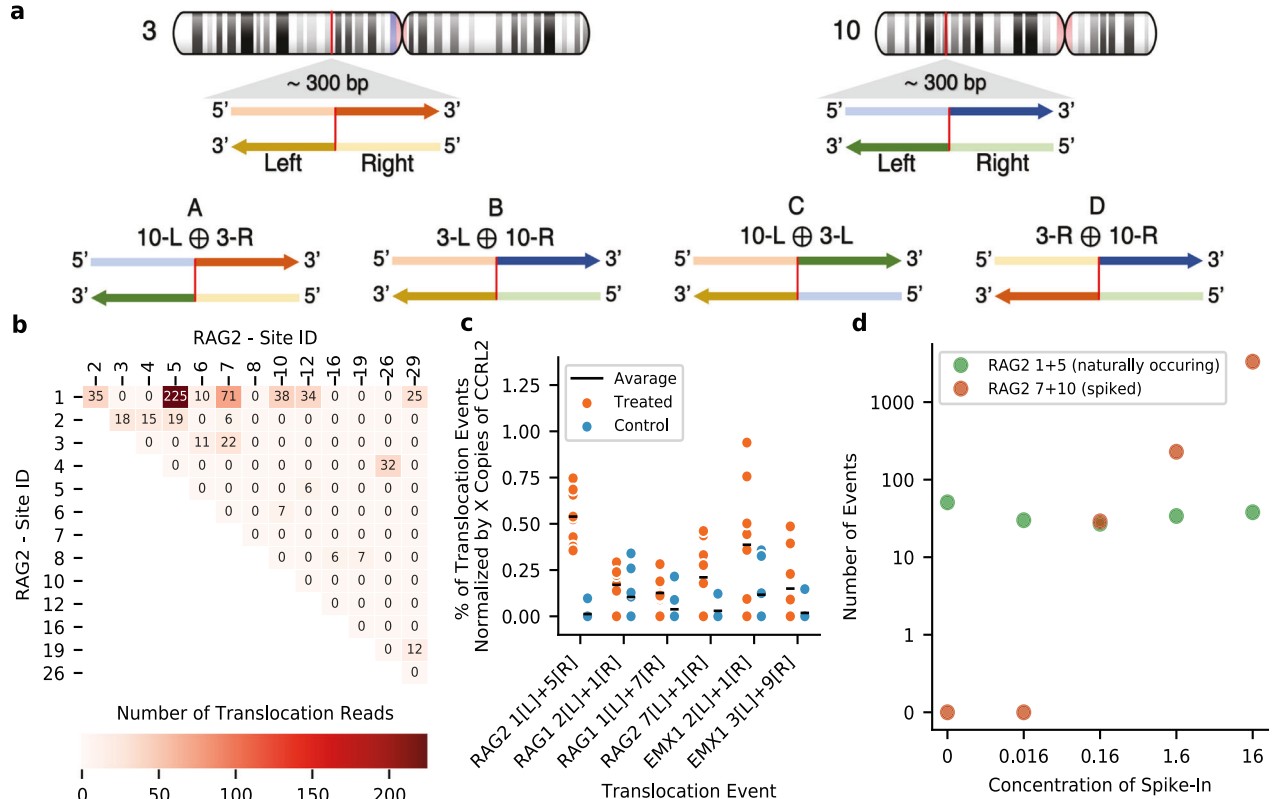

**Fig. 3 CRISPECTOR detects and quantifies adverse translocation events. a** Schematic description of all four translocation types. Schematic description of all possible translocations for chromosomes 3 and 10 p arms. The expected cut-sites are colored red and ⊕ denotes the concatenation/fusion of the sequences. The four possible fusion types give rise to structures that are either single-centromeric (A & B), centromere-free (C), or double-centromeric (D). **b** Translocation reads heatmap for *RAG2*, XT 2p, HEK293-Cas9. 19 translocations with a Hypergeometric (FDR corrected) *p*-value < 0.05 are presented in the heatmap with the associated read counts. For example: 225 reads were found between *RAG2_5* and *RAG2_1*, yielding a *p*-value of $1.53 \times 10^{-65}$. **c** ddPCR Validation of translocation events. Specific translocation or structural variation events were individually experimentally measured using event-specific ddPCR primers and probes. The means of 8 technical repeats are also depicted above (black lines). WRS (Wilcoxon Rank Sum)[41] one-sided *p*-values to support the higher Tx values measured are: $7.7 \times 10^{-4}$, 0.29, 0.027, 0.074, 0.09, and 0.17, ordered as in the figure. Source data is provided as a source data file. **d** Validation of CRISPECTOR translocation detection process based on synthetic spike-in standards. Edited *RAG2*, XT 2p, HEK293-Cas9 genomic DNA was spiked-in with serial dilutions (0, 0.016, 0.16, 1.6 and 16%) of *RAG2_10/RAG2_7* synthetic fusion construct (see text). Observed numbers of fusion reads for the spiked-in *RAG2_10/RAG2_7* construct (in orange) and for the naturally occurring *RAG2_1/RAG2_5* (in green) are presented.

genomic DNA) in both methods, which validates that CRISPECTOR does not report spurious events in this case. We note that the most frequent *RAG2* structural variation (*RAG2_1/ RAG2_5*) read counts, resembles read counts of the synthetic construct at the 0.16% spike-in level. As stated above, this naturally occurring translocation maintains a constant signal (Fig. 3d).

## Discussion

In this study, we demonstrate that CRISPECTOR supports accurate detection, evaluation, and quantification of indel and translocation genome-editing activity from data derived from an assay performed on paired Tx vs M samples using multiplex PCR followed by NGS. As such, CRISPECTOR addresses the validation, quantification, and characterization of off-target activity in genomic sites pre-identified by unbiased discovery approaches such as GUIDE-seq[8]. Expanding the statistical approach underlying the CRISPECTOR inference mechanism to also address data that comes from other types of Tx vs M experiments or technologies, including unbiased assays, is currently under development.

We demonstrate that CRISPECTOR's off-target indel calling results are more accurate compared to existing tools. The demonstrated reduced FP and FN rates, especially for challenging

loci with low editing rates, are important in all CRISPR applications and most notably in the context of therapeutic applications. Furthermore, CRISPECTOR enables the detection of alternative cut-sites at off-target loci. Finally, CRISPECTOR also supports the analysis of multiplex PCR and NGS data in an approach that detects, with high sensitivity, adverse structural variations, and translocation events occurring in an editing experiment. This last feature makes CRISPECTOR unique compared to other state-of-the-art tools.

Several validation processes have been used in this study. Translocation events were, as described above, experimentally validated using an independent ddPCR assay. Additionally, CRISPECTOR's sensitivity in detecting translocation events was examined by titrating synthetic products. In this titration experiment, we show that CRISPECTOR could accurately and consistently detect translocation constructs. As for indel events, we apply three lines of validation. The first one is based on human examination of all individual reads at the events called, conducted by two independent scientists. The second one consists of establishing the fact that events classified as actual editing events manifest a consistent behavior when measured in different experimental efficiencies. We expect off-target editing rates to increase as the efficiency increases. We do not expect such behavior for random or noise events.

Indeed, we observe such concordance (or monotonicity) with a $p$-value $< 1.7 \times 10^{-3}$ (binomial null, see "Methods" and see Supplementary Note 6 for detailed results). Finally, we reversed the roles of the Tx and M data. Specifically—the same data are analyzed where all reads coming from the Tx samples are labeled as M and vice versa. In this reverse orientation, we classify 10 out of the 1161 instances examined as edit events, compared to 215 out of 1161 in the correct orientation.

CRISPECTOR takes a conservative approach, considering all of the Tx reads classified as edits to indeed count as edit events and avoiding the subtraction of the number of similar M reads as is done by other tools. Instead, M misaligned reads are more systematically accounted for in the classification process. CRISPECTOR's approach thus reduces the FN rate in off-target loci but may lead to differences in estimated on-target rates. Moreover, CRISPECTOR uses tunable parameters that can be specified to balance between FP and FN. Studies related to the determination of these tunable parameters can potentially lead to better performance.

We note that an imbalance of Tx vs M read numbers affects the quality of CRISPECTOR inference, as in any comparative estimation. Thus, we advise the users to design experiments with a balanced sequencing procedure in high read depth. We also note that CRISPECTOR is designed for multiplex-PCR experiments. Nonetheless, it can also be used for singleplex-PCR experiments.

An important feature of CRISPECTOR is that it allows the user to infer editing rates without the need to accurately identify the CRISPR cut-site, thereby allowing identification of alternative cut-sites compared to the expected cut-site at any candidate locus. Indeed, we observe several such cases in this study. Some tools such as CRISPResso2 use a narrow quantification window centered at the predicted cut-site to avoid measuring errors resulting from PCR and sequencing. However, these tools can miss real editing events taking place outside this window. As we discussed above, faulty identification of the off-target gRNA targeting, due to incorrect interpretation, can lead to wrong positioning of the narrow quantification window. Additionally, alternative read alignments can place real edit events away from the expected cut-site. Finally, real editing activity can occur outside of the expected narrow quantification window due to the existence of an alternative PAM sequence. CRISPECTOR bypasses the need to determine this narrow quantification window by incorporating different prior probabilities in a larger window for each position in the reference sequence, thereby providing a more robust approach to the event quantification.

In recent years, strides have been made in the field of gene editing regarding the unbiased characterization, screening, mapping, and quantification of off-target editing mediated by engineered nucleases such as CRISPR-Cas9. Strategies such as GUIDE-seq[8], CIRCLE-seq[9], SITE-seq[10], and DISCOVER-Seq[11] have been established as proven tools for identifying genome-wide off-target activity caused by CRISPR-Cas9 genome editing. CRISPECTOR is the first tool of its kind that facilitates the inference of adverse translocation events in CRISPR-Cas9 editing, based on general design Tx vs M multiplexed, target-specific PCR data. The PCR panel used for the experiment would typically be designed based on data from an unbiased off-target detection process, such as GUIDE-seq. Translocation detection techniques were previously reported in the literature, such as HTGTS[18] and UDiTas[19]. However, these techniques address a fixed predetermined list of potential events on one side. For example, in UDiTas genomic sequences are amplified by one sequence-specific primer, targeted to a specific genomic sequence, on one side, and a second tagmentation-mediated primer on the other side. Thus, this technique allows for identifying any edit events, including structural variations, as well as any translocation

partners of a specific off-target site, at a particular predetermined genomic location. As DSBs are important components of the process leading to the translocation, it is also important and informative to investigate all the possible pairwise translocation events that can take place within a given set of identified potential (e.g, by GUIDE-seq, Circle-Seq, etc.) off-target sites. CRISPECTOR allows, in principle, for a comprehensive evaluation of all possible translocations among the predicted off-target sites addressed by the multiplex-PCR design without performing any additional experiments. Utilizing CRISPECTOR to analyze results from the off-target screening quantification experiment will produce, in the same run, the detected translocation events. Furthermore, CRISPECTOR can potentially support the analysis of genome-editing experiments with multiple guides, which could result in a higher rate of translocations. Therefore, CRISPECTOR is unique in its support for detecting translocation events due to genome editing, irrespective to cell type, based on comparative multiplex-PCR data. CRISPECTOR can be extended, by design, to also cover fragile genomic loci.

One possible issue with interpreting the results of PCR-based editing activity measurement approaches is related to artifacts resulting from the amplification. Multiple reads that represent a nominally identical edit event may actually be representative of any number of such events in the genome, between 1 and the full observed read multiplicity. Consider, for example, the 18 reads with 4 bases deleted, as in Fig. 2. This can represent 18 unique molecules or amplification of a single event of this type (see Supplementary Note 11, for more detailed explanation). The use of unique molecular identifiers (UMIs) as part of the multiplex-PCR experiments may serve to offset amplification-related biases[35]. Future versions of CRISPECTOR will support the use of UMIs.

In this study, we have demonstrated adverse translocations in treated *RAG1*, *RAG2*, and *EMX1* loci in a HEK293-Cas9 cell line. Our findings were confirmed using ddPCR, an orthogonal experimental approach. Interestingly, we detected occurrences of all four possible configuration types of translocations for different pairs of loci including centromere-free, double-centromeric, and two single-centromeric configurations. Future CRISPECTOR versions will analyze and report translocations by types, and therefore, will allow deeper mechanistic studies of these adverse events. Adjustment of the experimental protocol can also contribute to the examination of more configurations. In particular, the NGS protocol used in this work uses a fixed matching between NGS adaptors and PCR primers as part of the design. Specifically, using a PCR panel with both P5 and P7 on both the reverse and forward primers facilitates measuring all four types of translocations at every potential fusion site covered by the assay. Clearly, our methods do not detect translocations formed by fusion at sites not covered by the PCR panel.

In conclusion, CRISPECTOR has the potential to significantly improve the accuracy of genome-editing measurements and to allow improved mechanistic insights into this process. It will thereby enhance and accelerate the sound and accurate broader use of genome editing in biotechnology and therapeutic applications.

## Methods

**Overview**. CRISPECTOR is a software package with an easy-to-use command-line interface. The tool accepts FASTQ files in the Tx-M setup and a configuration table describing the experiment setup, which contains on-target and off-target site IDs, gRNAs (gRNA target sequence), reference amplicons, and primers (Supplementary Fig. 1). A graphical report is generated to visualize multiple detected potential off-target editing activity. The graphical representation of the results includes FASTQ processing information, distributions of indels as detected in Tx and M, the decisions of the classifier as applied to reads and indel editing activity per locus. For translocations we report the reads supporting the findings and a summary of the

results. Reported results come with statistical confidence indicators. Figure 1c depicts the CRISPECTOR workflow.

As noted above, CRISPECTOR takes as input Tx and M FASTQ files as well as reference sequences for the loci of interest. The first step of the process consists of adapter trimming, merging pair-end reads, and filtering out low-quality reads. This is done using FASTP[36]. Second, all reads are assigned to a particular reference sequence, or locus of interest. This assignment is done by matching read prefixes to the expected primers that pertain to the different loci/amplicons (lowest edit distance). Assignment is then verified by aligning the read suffix to the other primer (Supplementary Note 7). Some reads, for which no verified matching is obtained are moved to a second line of analysis. These are either artifacts or they may be originating from translocation events. Finally, to align assigned reads to the locus from which they originated we used an optimized version of the Needleman-Wunch algorithm[37] (Supplementary Note 8).

At the end of the pre-processing step we have, for each locus of interest, a list of aligned reads coming from the M experiment, denoted $L_M$ and a list of aligned reads coming from the Tx experiment, denoted $L_{Tx}$. The comparison of these two lists is the basis for quantifying the activity detected at the loci of interest.

Determining NHEJ/indel CRISPR activity at a given locus $\lambda$, is performed as follows:

1. Assign all reads into indel types per position. This results in lists of reads $L_M(\tau, i)$ and $L_{Tx}(\tau, i)$, where $\tau$ denotes the indel type, e.g. insertion of length 3, and $i$ denotes the index of the position at the given reference sequence. Note that reads that are perfectly aligned to the genomic reference, are not members of any of the above lists. For off-target sites these are typically a majority. Also note that reads may represent more than one indel implicated by the alignment.
2. For any $(\tau, i)$ we use a classifier (see details below) to determine whether reads that represent this indel $\tau$ at position $i$, originate from an editing event, or from background noise.
3. We count all the $L_{Tx}$ reads that are classified as originating from an edit event (in at least one position) and calculate their proportion from the total reads at $\lambda$, as well as a confidence interval for that proportion.

As noted in the Introduction, an important contribution of the CRISPECTOR pipeline is that it facilitates the detection of translocation events with fusions at on-target and off-target cut-sites.

As we worked with rhAmpSeq data and since the multiplex-PCR reaction contains all primer pairs it is possible that amplicons can be formed based on fusion molecules, as the primers on both sides will be present (Fig. 3a). Determining translocation activity between two loci $\lambda_1$ and $\lambda_2$, based on this observation, is performed as follows (also see Supplementary Fig. 8):

1. Identify all reads that are putatively originating from translocations.
2. Assuming that such reads are found for the pair $(\lambda_1, \lambda_2)$ construct the putative amplicon that represents the $\lambda_1$ to $\lambda_2$ fusion at the cut-site (as determined from the PAM).
3. Perform an alignment of all reads from Step 1, to the amplicon formed in Step 2. Retain reads with sufficiently strong alignment scores (details in Supplementary Note 8). Note that the above steps are performed for both Tx and M.
   We now have counts for reads attesting to fusion for both Tx and M: $C_{Tx}(\lambda_1, \lambda_2)$ and $C_M(\lambda_1, \lambda_2)$.
4. Using the above counts and the total number of relevant reads, in both Tx and M, we apply a hypergeometric test (see details below) to determine whether a translocation is likely to have occurred, as well as associate a $p$-value to this call.

**Classifying reads as originating from NHEJ/indel editing activity**. We aim to distinguish reads coming from CRISPR-edited molecules from reads derived from various sources of noise. This distinction is based on the ability to assess whether a specific indel originates from a CRISPR edit event or is a result of experimental background noise (such as sequencing artifacts or erroneous read assignments). We developed an algorithm for identifying edited reads, based on the comparison of indel statistics in the Tx-M setup (See algorithm overview in Supplementary Fig. 13).

First, for each locus in the Tx and M read lists, $L_{Tx}$ and $L_M$, are converted into several more focused lists pertaining to specific indel types and positions, $L_{Tx}(\tau, i)$ and $L_M(\tau, i)$. For a given indel type (and a given locus) we can summarize the information in the form of a table where each column represents a position on the reference locus (as illustrated in Supplementary Fig. 13). CRISPECTOR covers a broad range of possible indels (Supplementary Note 9).

Second, for each indel type, $\tau$, CRISPECTOR applies a Bayesian inference classifier on every position in the amplicon reference. The $\tau$ indel events observed in each such position, within the Tx reads, are classified as originating from an edit event or from background noise. All Tx reads with $\tau$ indels in position $i$, which were positively classified (edit event), are considered as edited reads. This process is repeated for all $(\tau, i)$ pairs. Eventually, all reads that were classified as positive in at least one pair $(\tau, i)$ are marked as edited reads. Because we expect an indel that originated from an edit event to be a result of a double-strand break at the cut-site,

only a window around the expected cut-site is considered for classification. All other positions are discarded (in our demonstration data window size is 20 bp).

The Bayesian classifier is a MAP (*Maximum A Posteriori*) estimator. Given an indel type and a reference position, $(\tau, i)$—let $n_{Tx} = |L_{Tx}(\tau, i)|$ and $n_M = |L_M(\tau, i)|$. Given these observed numbers, we can ask if the observation is more likely to have originated from an edit event—$P(edit | n_{Tx}, n_M)$, or to represent background noise—$P(no\ edit | n_{Tx}, n_M)$. MAP classifies a position as originating from an edit event when the posterior $P(edit | n_{Tx}, n_M)$ is higher.

MAP estimator probabilities are compared using Bayes Rule:

$$(1) \quad P(edit | n_{Tx}, n_M) > P(no\ edit | n_{Tx}, n_M) \iff$$
$$P(edit) \cdot P(n_{Tx}, n_M | edit) > P(no\ edit) \cdot P(n_{Tx}, n_M | no\ edit)$$

Where $P(edit)$ and $P(no\ edit)$ are the *Prior Probabilities*. Due to the lack of experimental annotated data in the field of genome editing, we define our priors based on the basic principles of CRISPR activity. These priors are configurable. In our demonstration data and experiments, $P(edit)$ is set to 0.5 at the expected cut-site position. For off cut-site positions we use smaller values in a decreasing order, relative to distance from the cut-site ($10^{-1} \rightarrow 10^{-4}$). Note, that non-zero *Prior Probabilities* away from the cut-site allow for the detection of alternative cut-sites in the classification window and are fully user-configurable.

$P(n_{Tx}, n_M | no\ edit)$ is the probability of the observation given the no edit model. Namely, the probability of seeing the observed number of Tx indels, $n_{Tx}$, out of the total number of indel events, $n_{Tx} + n_M$, under the no-edit assumption. This model means that all indels are equally likely to occur in a M read as in a Tx read. The random variable $n_{Tx}, n_M | no\ edit$ is modeled using the hypergeometric distribution. Let $N_{Tx} = |L_{Tx}|$ and $N_M = |L_M|$ denote the total number of reads in Tx and M, respectively. Without loss of generality (due to the symmetry of the hypergeometric distribution), we define $X | no\ edit \sim HG(N, B, n)$, where $N = N_{Tx} + N_M$, $B = n_{Tx} + n_M$, $n = N_{Tx}$ and so $P(n_{Tx}, n_M | no\ edit) = P(X = b) = HG(b; N, B, n) = \dfrac{\binom{B}{b}\binom{N-B}{n-b}}{\binom{N}{n}}$, where $b = n_{Tx}$. We thereby get:

$$(2) \quad P(n_{Tx}, n_M | no\ edit) = \frac{\binom{n_{Tx} + n_M}{n_{Tx}} \binom{N_{Tx} + N_M - n_{Tx} - n_M}{N_{Tx} - n_{Tx}}}{\binom{N_{Tx} + N_M}{N_{Tx}}}$$

$P(n_{Tx}, n_M | edit)$ is the probability of the observation given the edit model. Our model is based on the assumption that in a position and type where edit actually occurs, most observed indels are caused by an edit event, and a small portion is due to background noise. $P(n_{Tx}, n_M | edit)$ is the probability of seeing the observed number of Tx indels, $n_{Tx}$, out of the total number of indel events, $n_{Tx} + n_M$. Let $q$ be the probability of an indel to occur through an edit event (as noted above, a number close to 1). The parameter $q$ is inferred from the experimental data (Supplementary Note 10), and then the random variable $n_{Tx}, n_M | edit$ is modeled with the Binomial distribution. Without loss of generality, we can define $X | edit \sim Binom(n, q)$, where $n = n_{Tx} + n_M$, and so $P(n_{Tx}, n_M | edit) = P(X = k) = \binom{n}{k} q^k (1-q)^{n-k}$, where $k = n_{Tx}$. We thereby get:

$$(3) \quad P(n_{Tx}, n_M | edit) = \binom{n_{Tx} + n_M}{n_{Tx}} q^{n_{Tx}} (1-q)^{n_M}$$

**Quantifying indel editing activity**. The final editing activity is determined as the frequency of the edited reads out of the total number of reads in Tx. Note that this is a conservative approach (see "Discussion"). A confidence interval is calculated for each potential target site using the standard approach:

$$(4) \quad CI = \hat{p} \pm \Phi^{-1}\left(1 - \frac{\alpha}{2}\right) * \sqrt{\frac{\hat{p}(1-\hat{p})}{N_{Tx}}}$$

where $\hat{p}$ denotes the inferred editing frequency and $N_{Tx}$ denotes the total number of reads in Tx. $\alpha$ is the desired confidence level (which is 0.05 in our demonstration data) and $\Phi$ is the CDF of the standard normal distribution.

**Translocations**. Reads that are putatively originating from translocations are detected and collected in the demultiplexing step. A read $r$ is considered as a putative translocation read if one of the following holds (Fig. 3a depicts all four translocation types):

1. $Pfx(r) \unrhd F(\lambda_1) \wedge Sfx(r) \unrhd Rev(R(\lambda_2))$. Or, its reverse-complement: $Pfx(r) \unrhd R(\lambda_2) \wedge Sfx(r) \unrhd Rev(F(\lambda_1))$.
2. $Pfx(r) \unrhd F(\lambda_2) \wedge Sfx(r) \unrhd Rev(R(\lambda_1))$. Or, its reverse-complement: $Pfx(r) \unrhd R(\lambda_1) \wedge Sfx(r) \unrhd Rev(F(\lambda_2))$.
3. $Pfx(r) \unrhd F(\lambda_2) \wedge Sfx(r) \unrhd Rev(F(\lambda_1))$. Or, its reverse-complement: $Pfx(r) \unrhd F(\lambda_1) \wedge Sfx(r) \unrhd Rev(F(\lambda_2))$.
4. $Pfx(r) \unrhd R(\lambda_2) \wedge Sfx(r) \unrhd Rev(R(\lambda_1))$. Or, its reverse-complement: $Pfx(r) \unrhd R(\lambda_1) \wedge Sfx(r) \unrhd Rev(R(\lambda_2))$.

Where $F(\lambda_j)$ and $R(\lambda_j)$ are the forward primer and reverse primer of site $j$, respectively. $Pfx(r)$ and $Sfx(r)$ are $r$ prefix and suffix. $Rev()$ denotes the reverse-complement function of a nucleic acid sequence and $\unrhd$ denotes the best match

(lowest edit distance), for a given partial read, within the total set of primers and their reverse complements.

The identification of such reads is done for both Tx and M. We now have counts for reading attesting to fusion for both Tx and M: $C_{Tx}(\lambda_1, \lambda_2)$ and $C_M(\lambda_1, \lambda_2)$. Similar to indel editing activity estimation, translocations are evaluated from a noisy background. Erroneous read matching, un-stringent quality filtering, or guide mediated primer-dimer effects could cause false assignment of translocations. We note that we also observe such formations in the validation experiments (including ddPCR). To address the above issues, CRISPECTOR performs a hypergeometric test (Eq 5) for all pairs of active target sites (those sites investigated with the rhAmpSeq assay at which an estimated indel editing activity is higher than a configurable threshold - 0.1% in our demonstration data/experiments). The test is based on the numbers $C_{Tx}(\lambda_1, \lambda_2)$ and $C_M(\lambda_1, \lambda_2)$ calculated above.

Consider possible translocations between target sites $\lambda_1$ and $\lambda_2$. Let $b = C_{Tx}(\lambda_1, \lambda_2)$, $B = C_{Tx}(\lambda_1, \lambda_2) + C_M(\lambda_1, \lambda_2)$. For $n$ we take the geometric average of the total number of Tx reads in both sites, and $N$ the geometric average of the total number of reads (Tx and M) between the two sites. $n = \sqrt{C_{Tx}(\lambda_1) \cdot C_{Tx}(\lambda_2)}$ and $N = \sqrt{C_{Tx}(\lambda_1) \cdot C_{Tx}(\lambda_2)} + \sqrt{C_M(\lambda_1) \cdot C_M(\lambda_2)}$.

The hypergeometric tail probability of our observation is:

$$(5) \quad HGT(b; N, B, n) = \sum_{i=b}^{\min(n,B)} \frac{\binom{B}{i}\binom{N-B}{n-i}}{\binom{N}{n}}$$

The list of p-values thus obtained for all considered pairs is then FDR[38] corrected. All translocations with FDR below a configurable threshold (0.05 in our experiments) are reported in a table of reads, and aggregated results for the entire experiment are provided in a table format (Supplementary Fig. 9) and a heatmap plot (as in Fig. 3b). These representations of the results are included in the standard CRISPECTOR html report. Supplementary Fig. 8 depicts an overview of the algorithm.

**Concordance of activity rates**. We consider four pairs of experiments in EMX1, as detailed in Supplementary Note 6. In each one of these pairs, the first experimental condition is less conducive to genome editing than the subsequent one. This is evidenced by the on-target activity measured therein. For example: in EMX1_XT_2p_1_1.2_no_EE_4uM_CD34Plus there is less editing than in EMX1_XT_2p_1_1.2_EE_4uM_CD34Plus and. Indeed, the estimated editing activity of the on-target site of the first is 9.4%, compared to 73.31% of the latter. Now consider all the potential EMX1 off-target sites tested. For a locus where actual activity occurs we expect the levels detected to be consistent with the on-target monotonicity. For a locus where there is no actual editing activity we expect a random 0.5/0.5 behavior w.r.t this concordance. To support the validity of our activity calls we therefore consider 13 potential instances, with respect to the experiments listed in Supplementary Note 6, where non-zero activity is reported by CRISPECTOR for the stronger experiment. For each one of these, we have two reported activity levels. Our null hypothesis will be that the number of instances for which these two levels are concordant (monotone) with the on-target levels has a $Binom(n = 13, p = 0.5)$ distribution.

**CRISPR editing experiments and samples**. The data from a total of 28 experiments were used in this paper (Supplementary Table 1). The sequencing data of RAG1 and RAG2 editing in human CD34 + hematopoietic stem and progenitor cells (HSPCs) was obtained from Shapiro et al., 2020[12], sequence read archive (SRA) accession: PRJNA628100. The HPRT and AR sequencing data was obtained from Vakulskas et al., 2018, SRA accession: SRP150376. For an extra validation, additional experiment sets for RAG1, RAG2, and EMX1 in HEK293, HEK293-Cas9, and human CD34 + HSPCs, were conducted, and deposited to the SRA, under accession number: PRJNA630002.

**gRNA synthesis**. gRNAs were synthesized by Integrated DNA Technologies (IDT, Coralville, IA). The 20bp-long target-specific sequences were as follows: RAG2: 5′-TGAGAAGCCTGGCTGAATTA-3′; RAG1: 5′-AACTGAGTCCCAAGGTGGGT-3′; AR: 5′-GTTGGAGCATCTGAGTCCAG-3′; HPRT: 5′-AATTATGGGGAT TACTAGGA-3′; and EMX1: 5′-TTCTTCTTCTGCTCGGACTC-3′. Four gRNAs formulations were used: in vitro transcribed sgRNA (IVT) and three chemically modified gRNAs: Alt-R single guide RNA (sgRNA) and two forms of 2-part gRNAs, composed of either Alt-R crRNA (crRNA 2p) or Alt-R crRNA XT (XT 2p) annealed to a trans-activating CRISPR RNA molecule (tracrRNA). The XT 2p gRNA is bearing an additional modification in the crRNA, to further increase its stability. 2-part gRNAs and sgRNAs were resuspended in 1× Tris-EDTA (TE), PH 8, solution (IDT, Coralville, IA) to a concentration of 200 and 100 μM, respectively.

**gRNAs and RNP complex preparation**. Alt-R crRNAs and Alt-R tracrRNA (IDT, Coralville, IA) were mixed in an equimolar ratio, heated to 95 °C for 5 min, and then annealed at room temperature for 10 min to allow the formation of crRNA/tracrRNA duplexes. For each electroporation reaction, 104 pmol of Alt-R WT or HiFi Cas9 nuclease (IDT, Coralville, IA) were complexed with 260 pmol or 120

pmol gRNA (1:2.5 or 1:1.2 Cas9:gRNA molar ratio, respectively). RNP complexes were formed during 10–20 min incubation at room temperature before the electroporation.

**Genome editing in mobilized CD34+ HSPCs and in HEK293/HEK293-Cas9 cell lines**. Human CD34+ HSPCs, derived from mobilized peripheral blood (MPB) (AllCells, Alameda, CA), were thawed and cultured for 48 h, at a density of $2.5 \times 10^5$ cells/ml in StemSpan SFEM II (Stemcell Technologies, Vancouver, Canada) supplemented with: 1% penicillin/streptomycin (Biological Industries, Beit Haemek, Israel), SCF, TPO, Flt3-Ligand and IL-6 (100 ng/ml each. PeproTech, Rocky Hill, NJ). MPB CD34 + HSPCs were incubated at 37 °C, 5% $CO_2$, and 5% $O_2$ for 2 days prior to electroporation. HEK293 and HEK293-Cas9 were obtained from IDT (IDT, Coralville, IA) and were cultured in DMEM supplemented with 10% FBS, 0.1 mM NEAA, 6mM L-Glutamine, 1 mM Sodium Pyruvate, and 1% Penicillin/Streptomycin (Biological Industries, Beit Haemek, Israel). For HEK293-Cas9 cells, 500 μg mL$^{-1}$ G418 (Gibco Invitrogen, Grand Island, NY) was added for Neomycin-resistant cell selection. Cells were incubated at 37 °C and 5% CO2 and were allowed to grow to 70%-80% cell confluency. Prior to electroporation, MPB CD34+ HSPCs and HEK293 cell lines were reconstituted in P3 Primary Cell and SF Cell line electroporation solutions, respectively, according to the manufacturer's protocol (Lonza, Basel, Switzerland).

Both MPB CD34+ HSPCs and HEK293 cells were electroporated with RNP complexes at a final concentration of 4 μM. Most RNP reactions composed of a 1:1.2 of Cas9: gRNA molar ratio, were performed in the presence of 3.85 μM Alt-R Cas9 Electroporation Enhancer (EE, IDT, Coralville, IA). HEK293-Cas9 cells were electroporated with XT 2p gRNA at a final concentration of 10 μM. The cells were placed in the Lonza 4D-Nucleofector and electroporated using the DZ-100 program for MPB CD34 + HSPCs and the CM-130 program for HEK293 cell lines. The recovered cells were cultured for 48–72 h prior to gDNA extraction using the QuickExtract solution (Lucigen Corporation, Middleton, WI).

**On- and off-target editing frequency quantification by rhAmpSeq**. The potential off-target sites for EMX1, RAG1, and RAG2 gRNAs were previously identified by GUIDE-seq[8,12]. The rhAmpSeq multiplex amplicon sequencing technology (IDT, Coralville, IA) was used to further quantify on- and off-target editing activity[34,39]. The accuracy of the multiplex rhAmpSeq technology is based on blockage primers containing RNA bases at the 3′ end of the primer, causing DNA/RNA hybridization. A perfect DNA/RNA alignment is cleaved by RNase H2, allowing continuance amplification. rhAmpSeq primers were designed to flank each off-target cut-sites identified by GUIDE-seq and pooled together for multiplex assay amplification (IDT, Coralville, IA). 20 ng DNA from each sample was submitted to a two-round rhAmpSeq PCR. rhAmpSeq amplicons were purified and sequenced using the Illumina MiSeq platform (v2 chemistry, 150-bp paired-end reads).

**Translocation validation**. We first used standard PCR amplification to validate the translocation events detected by CRISPECTOR. The most prevalent structural variation event amongst RAG2 off-target sites was RAG2_1/RAG2_5. Therefore, PCR reaction was conducted to amplify ~1000 bp amplicon flanking the ends of the aberrant resection (see Supplementary Table 3 for primer sequences). Next, we used the Bio-Rad QX200 ddPCR system (Bio-Rad Laboratory, Hercules, CAL) to verify the following translocation events: RAG1_1/RAG1_7, RAG1_2/RAG1_1, RAG2_1/RAG2_5, RAG2_1/RAG2_5, RAG2_7/RAG2_1, EMX1_2/ EMX1_1, EMX1_3/ EMX1_9, and RAG2_10/RAG2_7. Each ddPCR reaction mix contained a PrimeTime® Standard qPCR Assay with a set of target-specific primers and a FAM-labeled probe for a given translocation event, as well as a reference PrimeTime® Standard qPCR Assay with HEX-labeled probe for the CCRL2 gene and a set of target-specific primers. The probe/primer mix was synthesized at a 1:3.6 ratio (see Supplementary Table 3 for primer sequences, and unprocessed gel image in the source data file). Each reaction mix contained the following components: 2X ddPCR Supermix for Probes No dUTP (Bio-Rad Laboratory, Hercules, CAL), 1X of each PrimeTime® Standard qPCR Assay (900 nM primers and 250 nM probe, IDT, Coralville, IA), 1X FastDigest HindIII (Thermo Fisher Scientific, Waltham, MA), 10X FastDigest Buffer (Thermo Fisher Scientific, Waltham, MA) and 20–100 ng genomic template DNA, supplemented for a total volume of 20 μl with Nuclease-free water. The genomic DNA samples were fractionated into lipid droplets, according to the manufacturer's protocol (Bio-Rad Laboratory, Hercules, CAL), and transferred to 96-well plate for PCR amplification. The reactions were amplified using Bio-Rad PCR thermocycler (Bio-Rad Laboratory, Hercules, CAL) with the following program: 1 cycle at 95 °C for 10 min, followed by 40 cycles at 95 °C for 30 s and 55 °C for 3 min, and followed by 1 cycle at 98 °C for 10 min at a ramp rate of 2.2 °C/s for all steps. Subsequent to the PCR reaction, the 96-well plate was loaded in the QX200 droplet reader. The droplets from each well were analyzed and the concentration (copies/μl) of the translocation (FAM) and wild-type CCRL2 (HEX) copies were calculated using the QuantaSoft analysis software (Bio-Rad Laboratory, Hercules, CAL). The percentage of translocation events normalized to the number of CCRL2 copies was calculated as follows: $\left( \frac{\sum \frac{Copies}{\mu L} FAM}{\sum \frac{Copies}{\mu L} HEX} * 100 \right)$. Each assay was conducted in 8 replicates. The ddPCR source data is provided as a source data file.

**Synthetic translocation titration**. The *RAG2*_10/*RAG2*_7 synthetic construct was produced as MiniGene plasmid (IDT, Coralville, IA), representing an artificial translocation event between *RAG2* off-target site 7 and 10, in the form of P7 adaptor-5′-*RAG2*_10-*RAG2*_7-3′-P5 adaptor. Serial dilutions of the construct were performed and spiked into *RAG2*, XT 2p, HEK293-Cas9 genomic DNA in varying levels, ranging from 0.016-16%. The *RAG2*, XT 2p, HEK293-Cas9 genomic DNA and the spike-ins samples were subjected to ddPCR quantification and two-round rhAmpSeq PCR. The rhAmpSeq and ddPCR assays were performed as described above.

**Reporting summary**. Further information on research design is available in the Nature Research Reporting Summary linked to this article.

## Data availability
The main data supporting the findings of this study are available within the paper and its Supplementary Information files. All sequencing data generated in this study have been deposited in the National Center for Biotechnology Information (NCBI) under SRA accession: PRJNA630002. The sequencing data from Shapiro et al.[12], was obtained from the NCBI under SRA accession: PRJNA628100. The sequencing data from Vakulskas et al.[13], was obtained from the NCBI under SRA accession: SRP150376. The raw data underlying Fig. 3c as well as Supplementary Fig. 11, are provided as a source data file. All other relevant data are available from the corresponding author upon request. Source data are provided with this paper.

## Code availability
CRISPECTOR command line tool is available as a Bioconda package and as a Docker image at http://bioconda.github.io/recipes/crispector/README.html. The source code and the installation and user manual are available online at https://github.com/YakhiniGroup/crispector (https://doi.org/10.5281/zenodo.4561518)[40].

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

## Acknowledgements
The authors thank Chihong Choi, from IDC Herzliya, for leading the development of the html report. We also thank Mark Behlke from IDT for useful early discussions. We thank the Yakhini Research Group, at IDC and the Technion, and the Hendel Lab, in BIU, for numerous important discussions and comments. The Hendel Lab gratefully acknowledges the funding support from the European Research Council (ERC) under the Horizon 2020 research and innovation program (Grant agreement No. 755758). L. Anavy is supported by the Adams Fellowships Program of the Israel Academy of Sciences and Humanities.

## Author contributions
I.A., A.L.J., L.A., A.H., and Z.Y. conceived the project and planned the approach. I.A. and Z.Y. developed the statistics and the algorithmics with contributions from A.L.J. and L.A. I.A. developed all components of the software. M.S.M., G.K., Y.W., A.L.J., D.B., and L.A. contributed to testing and method improvement. O.I. led all the experimental work with contribution from D.A. and N.B.H. A.H. supervised the experimental work, with contributions

from GRR. L.A. participated in the project supervision. A.H. and Z.Y. jointly supervised this work. I.A., A.H., and Z.Y. wrote the manuscript, with contributions from all authors.

## Competing interests

G.K., M.S.M., and G.R.R. are employees of Integrated DNA Technologies (IDT), which sells reagents similar to some described in the manuscript. All other authors declare no competing interests.
