## [Peer Review File · Nature Communications]

Reviewers' Comments:

Reviewer #1:

Remarks to the Author:

The work by Amit et al., describe a novel software tool, CRISPECTOR, to evaluate genome-editing activity of RNA driven nucleases from multiplex targeted PCR and next-generation sequencing experiments using paired treatment/control experiments. This software tool uses a novel statistical approach at analyzing cutting efficiency from sgRNAs in CRISPR experiments by applying Bayesian statistics on mock and true experiments at known on- and off-target sites. CRISPECTOR uses a confidence interval to infer off-target activity rates of genome-editing technology. The evaluation of CRISPECTOR was done using five different sgRNAs. The implementation of a confidence interval to evaluate genome-editing efficiency provides a novel approach to the analysis of next-generation sequencing obtained from genome-editing experiments. CRISPECTOR also provides a tool to evaluate translocations from multiplex PCR experiments. Overall, the software tool and statistical analysis stands as an improved alternative approach to current tools to analyze data from genome-editing experiments. The description of the tool is complete; however, the manuscript lacks fluidity with regard to describing key aspects of the statistical analysis on supplemental figures. Part of the validation of CRISPECTOR is described as an "expert analysis", which lack enough information and a clear description of the thought process. A major gap is the absence of any information of how technical replicate sequences arising from PCR are filtered out in the analysis, which considerably affect the interpretation of the data. While of potential value to the field, the authors are urged to more clearly and succinctly describe the methodology.

MAJOR POINTS:

1. CRISPECTOR is portrayed as a novel tool for the detection of off-target activity. However, the tool depends on pre-determined/predicted off-target loci to generate the data from multiplex-PCR experiments. Even though the software analyzes multiple PAM sequences at a given off-target locus, the description of the tool to detect off-target and genome-editing associated translocations should be further clarified so as not to confuse with detection of robust previously unknown sites (using GUIDE-Seq here).
2. As this manuscript reads more like a highly technical methods paper, the manuscript lacks an explanation of the limitations of the technology that are linked to the use of multiplex-PCR to generate the data. The manuscript needs to include the possible limitations of the statistical analysis to be used in data sets different from multiplex-PCR.
3. There is no description on the filtering of technical duplicate events arising from PCR. In multiple analyzed data, i.e. the provided excel files in the supplementary information:
 - a) In Figure 2, the sequence reads being analyzed present a pattern characteristic of technical duplicate reads rather than biological duplicates. There are 18 reads with 4 bases deleted and/or 9 reads with 10 bases deleted exactly at the same place and no other reads with >6 bases deleted anywhere else. The number of reads is at odds with the fact that there are no events with 3-4-5 or more deletions and suggestive of low sequence diversity with respect to the sequencing depth.
 - b) In the data file RAG1_XT_2p_HEKCas9_RAG1_51 treatment_aligned_reads. There are 4 reads with 17 bases deleted: 6 reads generate the same sequence, and 3 reads have 1 mismatch further away from the deletion (a possible sequencing error, no information on the quality of the read was provided). Again, the pattern of the reads is more of a technical duplicate rather than a biological repeat.
 - c) 19 reads with 1 insertion: all "A", 8 reads with 34 exact same bases inserted at the same position.
4. The data presented in figures are generally difficult to assess as to the level of the type of replication. If noncutting controls are included in a given sequencing run (major source for sequence read error) this should be clearly stated.
5. The comparison of CRISPECTOR to CRISPResso2 and ampliCan tools. The comparison lacks the interpretation of CRISPResso2 in the reads in Supplementary Note S2. The authors simply indicate that errors exist in the analysis of CRISPResso2 by showing the reads without indicating the result

of CRISPResso2 analysis. Data from CRISPResso2 should be presented on the 19 reads that differed from CRISPECTOR. The comparison of CRISPECTOR and ampliCan needs examples of reads that had different results between the tools. Without a detailed analysis between the different data interpretation of ampliCan and CRISPECTOR readers will have a difficult time to evaluate whether CRISPECTOR is better than ampliCan. Do CRISPResso2 and ampliCan have a filter for technical repeat reads? How does the analysis of technical repeats differ from CRISPResso2 and ampliCan with the one used from CRISPECTOR? The data for the comparison between CRISPECTOR and CRISPResso2 is missing (file referenced: S2_expert_validated_dataset_results.xlsx).

6. CRISPECTOR has the capacity to analyze translocations from known off-target loci. However, the tool is unable to detect translocations to new off-target sites or endogenous unstable parts of the genome in an experiment. The authors narrative on the capacity of CRISPECTOR to detect events from both sides of a known on- and off-target site wrongly portrays the software tool as a novel method for the biological analysis of translocations (see also minor point 6). In this regard, CRISPECTOR could be described as a complementary analysis to other tools developed for the detection of previously unknown off-targets.

7. The validation of CRISPECTOR is described as "manual expert validation". To support this, a detailed explanation of the thought process behind the validation of sequence reads needs to be provided. For instance, the authors mentioned that the expert checked "all the possible alternative alignments". What was the analysis behind the validation? What are the limits covered by the statement "all possible alternative alignments"? In such validation analysis, if the TX data and Mock data are reverse, how does CRISPECTOR behave?

8. The discussion section should address limitations on the data collected from multiplex-PCR by stating the limitations of the technique. The authors should address the amount of work required for the adaptation of CRISPECTOR analysis to be used in next generation sequencing data obtained by different techniques.

9. In the discussion section, the authors should address the dependency of CRISPECTOR in assays that identified the off-target loci, particularly, the dependency on GUIDE-Seq. Such discussion should address the coverage depth of off-target sites and how are they selected from GUIDE-Seq data.

MINOR POINTS

1. Figure 2 needs substantial formatting.

a. Figure 2a is missing y-axis. It is apparent that the y-axis for each row is different. The legend needs to indicate the meaning of all the acronyms used in the figure e.g. Del 1, Del 2, Del 3. The circles should be highlighted in different color to help the viewer distinguish between the information presented. The gray boxes used to indicate edit events are too light, a suggestion is to use a different color.

b. Figure 2c has a typo on "the numbers of instances". The text should be "the number of instances". The sub-figure needs to use the space more efficiently, too much blank space in between schematics and the font number used inside the pie charts is very small.

2. Page 5, line 96 and 97. The authors do not acknowledge the existence of tools designed to detect genome-editing associated translocations ("in the context of this work" might need clarification). Moreover, the sentence syntax needs to be changed.

3. Page 7, line 161. The percentage indicated in the text do not match the percentage in the legend of Figure 2.

4. Page 12, line 257. Last sentence of the paragraph is missing a preposition.

5. Supplemental Figure 3 is not clear. Indel editing activity of what? Off-targets? Does n equal the number of events detected, the number of experiments performed on each off-target, or the number of off-targets? The legend does not indicate why red dots represent 0% (presumably those Off-targets with no discernable editing). Why does RAG1 have two sites with 0 edit distances?

6. With regard to the translocations detected overview in Supplemental Figure 7, does the hypergeometric test account for increased enrichment of intrachromosomal translocations due to increased interactions in cis? Further on the topic of translocations, are the authors taking into account the inability to sequence commonly primed (i.e. P5/P5 and P7/P7) illumina sequence tails

and thus underestimating the translocation frequency by as much as 2-fold? This should be discussed given that the authors displayed the detectable combinations in Supplemental Figure 9 but failed to note this caveat in their frequency calculations.

7. Supplemental Figure 9 has a typo in the legend.

8. Supplemental Table 2. The legend contains an error "Error! Reference source not found".

9. In Supplementary Note S2, "Furthermore, this step was taken to make sure that reads were not overlooked due to the primer-matching algorithm that is described in Supplementary Note 6.". The reference should be to Supplementary Note S7.

10. Line 612: transcription  translocation

11. Line 736: What is a WRS statistical analysis? It would appear based on the number of technical replicates presented that ddPCR really only confirmed the dominant translocation (may want to recheck p-values reported in legend), which is in contrast to the statement in line 242 (experimentally confirmed which sites?); the statement is vague and does not represent the actual findings.

Reviewer #2:

Remarks to the Author:

In therapeutic gene editing, NGS amplicon sequencing is the gold standard for analysis of editing at off-target sites. Amit and Iancu et al. develop a software package, termed CRISPECTOR, for analyzing NGS amplicon sequencing experiments for CRISPR editing, focused on multiplex experiments for off-target analysis that involve comparisons between treated and control cells. They demonstrate improved accuracy examining editing on 5 pooled multiplex-PCR NGS experiments compared to CRISPResso2 and ampliCan. This is primarily validated by human expert examination and the dose-dependence of indels. Because multiple sets of primers are used in multiplexed PCR, one set for each site, the authors are able to look for mixing of primers between primer sets in resulting reads which indicate translocations. They use this translocation "module" to detect translocations in a number of experiments, which are partially validated by ddPCR.

The authors describe very clearly the limitations of existing packages and what they hope to accomplish with this new package. The key innovations compared to existing packages are 1) inference of edit percentages with confidence intervals at putative cut sites using a Bayesian inference approach and 2) a method for translocation detection at putative cut sites by examining reads. The improved inference of editing appears to be primarily important when editing percentages are very low or background noise is very high, so its utility is limited. The translocation detection method could be helpful, although it requires a demonstration of sensitivity. Overall, the CRISPECTOR package may become a useful tool for the community, but a few points should be addressed with respect to the key claims:

Improved Inference of Editing:

Looking at the reads by hand, the examples highlighted in the paper all seem to be situations where even experts have to partially guess. I would have liked to see more than one expert validating these reads. Alternatively, blind expert validation where the expert is presented with unlabeled samples and told to identify whether editing occurred or not, or blinding where the expert is presented with pairs of reads, sometimes mock vs mock, would also suffice. The expert validation is not described in the methods.

In most cases where CRISPResso2 seems to differ, including all statistically significant instances in Figure 2b, CRISPResso2 underestimates the editing percentage, primarily due to missed long indels. Missed long indels also account for mistakes with an alternative cut site. If the quantification window size parameter is varied for CRISPResso2, what are the results? If using a

slightly larger window eliminates these discrepancies, then this is not a fair comparison. A varied parameter set is described for ampliCan but not for CRISPResso2 in Supplementary Note S6.

The text says "CRISPECTOR is designed to be highly sensitive, detecting events at rates as low as 0.1%. Such events occur quite often, as shown in Figure 1a and 1b." To me, this implies that CRISPECTOR is detecting editing rates as low as 0.1% accurately. However, Figure 1a and b instead show that, at certain cut sites, the indel rate for mock samples is sometimes higher than 0.1%. I would have expected to see a scatter plot of treatment vs. mock indel rate, a scatter plot of inferred editing rate vs. mock indel rate, or a histogram of inferred editing rate.

In the 19 sites of disagreement between CRISPResso2 and CRISPECTOR, seeing dose dependence (or the lack of) at these sites would help solidify the edited/unedited claim.

Minor comment: There now exist a number of algorithms/tools for indel prediction (FORECasT, inDelphi, Lindel, ...). Is it possible to incorporate the results from these tools as priors for $p(\text{edit})$ for a specific indel outcome or as an aggregate for a particular nucleotide position, varying by gRNA?

Minor comment: The estimated background noise α is already quite conservative by choosing the 95th percentile of $\alpha(j)$, but in general it seems like it should vary from cut site to cut site, although I'm not sure how to do this rigorously. What does the ROC curve look like if you vary the percentile chosen for α ? It's also not clear to me what "The final $q(j)$ is averaged with a configurable pre-defined probability (0.9 in our experiments), which encourages stable estimation." means.

Minor comment: $<20/3000$ reads is not something I would describe as a high rate in Supplementary Figure 4. Also, I assume 3 is supposed to be 13 in "18 of these 19 instances were determined, by expert investigation of the reads, to be CRISPResso2 errors – 5 false-positives and 3 false-negatives."

Translocation Detection

The statistical analysis done here analyzing reads with the hypergeometric test is sound. However, to be a useful tool, the experimental side of this translocation module needs to have sufficient sensitivity to detect translocations adequately. To my knowledge, no one has shown that rhAmp-seq or similar methods can do this.

A possible set of experiments to show this include titrating small amounts of a synthesized "translocation product" into a multiplexed PCR containing templates for the standard amplicons.

Some detected translocations are validated by ddPCR/regular PCR. I would have liked to see validation that translocations not detected by CRISPECTOR can not be found by ddPCR/regular PCR.

Along these lines, comparisons to existing methods would be good to include. LAM-HTGTS may be more sensitive due to the enzymatic blocking step eliminating amplicons without translocation, so UDiTaS may be the fairest comparison. UDiTaS can easily be augmented to include multiple primers in the first round of PCR, as mentioned in the discussion of the manuscript.

Minor comment: The authors state that GUIDE-seq can not be used to detect structural variant events. The original GUIDE-seq paper shows translocation detection in their AMP-seq add-on in Figure 5, detecting all 4 types of translocations between an on-target site and an off-target site.

Minor comment: When inferring editing activity, the package is able to detect alternative cut sites.

Is this alternative cut site information used in generating the template for read alignment for translocations? The methods were not clear on this point.

Minor comment: Can CRISPECTOR be quantitative beyond detection/existence of a particular translocation? It looks like there is a correlation between translocation read counts and translocation activity, but showing this definitively would strengthen the paper.

Minor comment: I think the heat-maps would show more useful information if they were displayed as a symmetric matrix, with everything under the diagonal being read counts for statistically significant translocations from mock samples, and everything above the diagonal being read counts from treatment samples.

Other minor comments:

XT 2p is mentioned in multiple figures in the main text as well as the supplement but is not described anywhere in the text. What is XT 2p? crRNA XT:tracrRNA duplex?

The tool itself has a nice user interface and is easy to use. One suggestion for improvement is to integrate the edited reads table directly into the main output of the HTML report (so you don't have to click on it), and to format the table such that reads with insertions are shown properly aligned to the target site. Examples of other tools where this is done include inDelphi and OutKnocker.

Point-by-point response to the reviewer's comments

We thank the reviewers for your interest in our work, positive feedback and criticisms. We have included additional experimental results to our study and have made changes to the manuscript to address the specific comments and concerns of the reviewers (marked in red in the manuscript).

We believe that these changes have improved our manuscript considerably and that they clarify the points raised by the reviewers. Thank you for your consideration of these revisions and your assistance during this process. We look forward to publishing our work in Nature Communication.

Sincerely,

Zohar Yakhini

Computer Science Department, Technion, Haifa, Israel

School of Computer Science, IDC Herzliya, Herzliya, Israel

Ayal Hendel

Life Science faculty, Bar Ilan University, Ramat-Gan, Israel

The following is a point-by-point response to the reviewer's comments. Our responses are in blue.

REVIEWER COMMENTS

Reviewer #1 (Remarks to the Author):

The work by Amit et al., describe a novel software tool, CRISPECTOR, to evaluate genome-editing activity of RNA driven nucleases from multiplex targeted PCR and next-generation sequencing experiments using paired treatment/control experiments. This software tool uses a novel statistical approach at analyzing cutting efficiency from sgRNAs in CRISPR experiments by applying Bayesian statistics on mock and true experiments at known on- and off-target sites. CRISPECTOR uses a confidence interval to infer off-target activity rates of genome-editing technology. The evaluation of CRISPECTOR was done using five different sgRNAs. The implementation of a confidence interval to evaluate genome-editing efficiency provides a novel approach to the analysis of next-generation sequencing obtained from genome-editing experiments. CRISPECTOR also provides a tool to evaluate translocations from multiplex PCR experiments. Overall, the software tool and statistical analysis stands as an improved alternative approach to current tools to analyze data from genome-editing experiments.

>> We thank the reviewer for the accurate summary and appreciation of the advancements presented in this manuscript.

The description of the tool is complete; however, the manuscript lacks fluidity with regard to describing key aspects of the statistical analysis on supplemental figures.

>> We agree that clarity can be improved. We added details to the descriptions of the supplemental figures- lines # 26-27, 48, 53, 58-64.

See, for example SF3, S6, S7. In particular:

- In SF3 – We referenced Kendall τ and the corresponding p-values.
- In SF6 – We added confidence intervals to CRISPECTOR editing activity estimations.
- In SF7 – We added details and clarified the relative difference analysis.

Part of the validation of CRISPECTOR is described as an “expert analysis”, which lacks enough information and a clear description of the thought process.

>> We thank the reviewer for this critical comment. We will address this point below (under our response to the major points).

A major gap is the absence of any information of how technical replicate sequences arising from PCR are filtered out in the analysis, which considerably affect the interpretation of the data.

>> Indeed – this confounder in the process should be addressed. We first note that PCR amp bias applies equally to the Tx and the mock, which serves to narrow this quantification gap. That said, the use of UMIs in the measurement assay could further offset much of this gap. We are planning to include support for UMIs in future versions of CRISPECTOR and in future protocols for measuring editing activity. We think, however, that this further development is beyond the scope of this manuscript. The PCR bias issue is not addressed by state of the art tools. The current state of the development of CRISPECTOR, as reflected in the manuscript, represents an improvement that will already greatly help the community, as an alternative approach with significant added functionality, as stated by the reviewer. The PCR bias, pointed out by the reviewer, is thus addressed in the discussion section of our new version- lines # 393-400.

While of potential value to the field, the authors are urged to more clearly and succinctly describe the methodology.

>> We accept the comment and address it as detailed below (under our response to the major points).

MAJOR POINTS:

1. CRISPECTOR is portrayed as a novel tool for the detection of off-target activity. However, the tool depends on pre-determined/predicted off-target loci to generate the data from multiplex-PCR experiments. Even though the software analyzes multiple PAM sequences at a given off-target locus, the description of the tool to detect off-target and genome-editing associated translocations should be further clarified so as

not to confuse with detection of robust previously unknown sites (using GUIDE-Seq here).

>> We thank the reviewer for this comment. We have emphasized in the revised manuscript, that CRISPECTOR serves to validate and quantify off-target activity detected and reported by unbiased approaches such as GUIDE-Seq. We have clarified these in the abstract (lines# 35-36), discussion (lines# 306-308, 367-373, 381-384), and in the methods sections (lines# 671-672, 676-678).

2. As this manuscript reads more like a highly technical methods paper, the manuscript lacks an explanation of the limitations of the technology that are linked to the use of multiplex-PCR to generate the data. The manuscript needs to include the possible limitations of the statistical analysis to be used in data sets different from multiplex-PCR.

>> In the discussion we added a clarification stating that we intend to expand usage to address other measurement protocols under the same or similar statistical framework. One major direction of expansion will be the analysis of GUIDE-Seq data. The adaptation of our Bayesian approach to GUIDE-Seq data is not straightforward-lines# 308-310, and 393-400.

3. There is no description on the filtering of technical duplicate events arising from PCR. In multiple analyzed data, i.e. the provided excel files in the supplementary information:

a) In Figure 2, the sequence reads being analyzed present a pattern characteristic of technical duplicate reads rather than biological duplicates. There are 18 reads with 4 bases deleted and/or 9 reads with 10 bases deleted exactly at the same place and no other reads with >6 bases deleted anywhere else. The number of reads is at odds with the fact that there are no events with 3-4-5 or more deletions and suggestive of low sequence diversity with respect to the sequencing depth.

b) In the data file RAG1_XT_2p_HEKCas9_RAG1_51 treatment_aligned_reads. There are 4 reads with 17 bases deleted: 6 reads generate the same sequence, and 3 reads have 1 mismatch further away from the deletion (a possible sequencing error, no information on the quality of the read was provided). Again, the pattern of the reads is more of a technical duplicate rather than a biological repeat.

c) 19 reads with 1 insertion: all "A", 8 reads with 34 exact same bases inserted at the same position.

>> We thank the reviewer for this important point and for their attention and efforts invested in investigating the details provided by the CRISPECTOR report. Our comments are as follows:

a. We are happy to see that our report design allows for the user to carefully interact with the data.

b. The replicated aligned reads present in this particular example can be PCR artifacts

but can also be the result of a defined CRISPR activity that has some physical advantage. It is reasonable to assume that certain indel configurations can be more likely the result of editing activity (PMID: 30033371, 27499295, and 24535568). Such physically advantageous patterns will thus yield more representative reads.

c. By using UMIs in the protocol we would be able to distinguish between naturally occurring high multiplicity events and PCR duplicates. As stated above we are working on incorporating this option into the process and the s/w, as part of the next version.

d. To take a conservative approach we could collapse the Mock reads into their unique version. This is a process that requires parameter fine tuning and that would have introduced more FPs. We analyzed this example, in Appendix 1: “Accounting for the effect of PCR duplication” (can be found at the end of our answers for reviewer #1). We are happy to include this assessment in the supplementary material, if the reviewer recommends doing so, to be referenced from the discussion section.

e. As we provide smooth access to the analysis details the user may want to filter out such events as a post analysis step.

f. We added to the discussion a paragraph on this topic. We include a specific pointer to the example highlighted in the review.

Lines # 393-400.

4. The data presented in figures are generally difficult to assess as to the level of the type of replication. If noncutting controls are included in a given sequencing run (major source for sequence read error) this should be clearly stated.

>> In all the experiments we have included non-cutting controls (referred to as the mock experiments). We have clarified this in the methods (lines# 486-491) and discussion sections (lines# 336-342).

5. The comparison of CRISPECTOR to CRISPResso2 and ampliCan tools. The comparison lacks the interpretation of CRISPResso2 in the reads in Supplementary Note S2. The authors simply indicate that errors exist in the analysis of CRISPResso2 by showing the reads without indicating the result of CRISPResso2 analysis. Data from CRISPResso2 should be presented on the 19 reads that differed from CRISPECTOR.

>> This added detail certainly improves the presentation. We added the details as requested in Supplemental Note S2.

The comparison of CRISPECTOR and ampliCan needs examples of reads that had different results between the tools. Without a detailed analysis between the different data interpretation of ampliCan and CRISPECTOR readers will have a difficult time to evaluate whether CRISPECTOR is better than ampliCan.

>> When running ampliCan on Site 51 of *RAG1*, XT 2p in HEK-CAS9, the ampliCan final report generation crashes. Furthermore, ampliCan doesn't provide a read level activity calling information to the user. Therefore, it is impossible to break the comparison process into read level comparisons. We also note that ampliCan results

deviate from both CRISPResso2 and CRISPECTOR as exemplified in Sup Figure 6. Therefore, it may be of lesser interest to perform a rigorous comparison.

Do CRISPResso2 and ampliCan have a filter for technical repeat reads? How does the analysis of technical repeats differ from CRISPResso2 and ampliCan with the one used from CRISPECTOR?

>> CS2 and AmpliCan do not address technical repeats in their current versions.

The data for the comparison between CRISPECTOR and CRISPResso2 is missing (file referenced: S2_expert_validated_dataset_results.xlsx).

>> Thank you for this comment. The file S2_expert_validated_dataset_results was possibly lost in the original submission process. We attached this file again.

6. CRISPECTOR has the capacity to analyze translocations from known off-target loci. However, the tool is unable to detect translocations to new off-target sites or endogenous unstable parts of the genome in an experiment. The authors narrative on the capacity of CRISPECTOR to detect events from both sides of a known on- and off-target site wrongly portrays the software tool as a novel method for the biological analysis of translocations (see also minor point 6). In this regard, CRISPECTOR could be described as a complementary analysis to other tools developed for the detection of previously unknown off-targets.

>> Indeed – CRISPECTOR will only detect translocations involving sites covered by the multiplex-PCR panel. The entire mpx-PCR approach is based on using information inferred from an unbiased experiment and is designed to provide better quantification for such inferred activity sites. We understand that this was not clear and provided a better explanation to clarify this point, in the abstract (lines# 33-36), and in the discussion (lines# 369-373, 381-384).

7. The validation of CRISPECTOR is described as “manual expert validation”. To support this, a detailed explanation of the thought process behind the validation of sequence reads needs to be provided. For instance, the authors mentioned that the expert checked “all the possible alternative alignments”. What was the analysis behind the validation? What are the limits covered by the statement “all possible alternative alignments”?

>> We thank the reviewer for this critical comment. We have included in the revised manuscript a detailed explanation of the “human expert validation” process. Please see Supplemental Note S2- “Human validated dataset for the task of active off-target site classification” lines # 489-605.

In such validation analysis, if the TX data and Mock data are reverse, how does CRISPECTOR behave?

>> Indeed – the reverse analysis is a good control process. We did report these results in the discussion section of the original submission. Following the reviewer’s comment, we added text and details to this part- lines# 331-334.

8. The discussion section should address limitations on the data collected from multiplex-PCR by stating the limitations of the technique.

>> The limitations of the PCR approach, including the focus on predetermined potential targets and the amplification bias are now addressed in several places in the manuscript. The discussion has been modified to even further explicitly emphasize this important point, e.g lines 336-342, 344-346, and 393-400.

The authors should address the amount of work required for the adaptation of CRISPECTOR analysis to be used in next generation sequencing data obtained by different techniques.

>> We added a sentence to the discussion (lines# 308-310). The statistical approach of CRISPECTOR can, indeed, be applied to results acquired by other technologies. We are currently working on expanding the scope of CRISPECTOR in this direction. The adaptation work is not trivial and we hope to describe progress in a future report.

9. In the discussion section, the authors should address the dependency of CRISPECTOR in assays that identified the off-target loci, particularly, the dependency on GUIDE-Seq. Such discussion should address the coverage depth of off-target sites and how are they selected from GUIDE-Seq data.

>> We thank the reviewer for this very important point. Indeed, since we are introducing an off-target quantification method, it is important to elaborate on the entire process, starting from the unbiased identification/detection of the off-target potential sites and all the way to the actual quantification. In this manuscript we mostly facilitate the latter. We have addressed this part in the revised manuscript. We believe that the current text reflects the process and the role of CRISPECTOR in several places. Lines# 188-197, 306-308, 369-373, and 381-384.

MINOR POINTS

1. Figure 2 needs substantial formatting.

a. Figure 2a is missing y-axis. It is apparent that the y-axis for each row is different. The legend needs to indicate the meaning of all the acronyms used in the figure e.g. Del 1, Del 2, Del 3. The circles should be highlighted in different color to help the viewer distinguish between the information presented. The gray boxes used to indicate edit events are to light, a suggestion is to use a different color.

>> The horizontal panels in Fig 2 include an indication of the numbers represented by the bars in lieu of a y-axis. This is our preferred presentation. We explained Del X as requested. We changed the focus ellipses according to the comment. We thank the

reviewer for the suggestion re the light grey background. We will consider a visual revision for the next version of the software package.

b. Figure 2c has a typo on “the numbers of instances”. The text should be “the number of instances”. The sub-figure needs to use the space more efficiently, too much blank space in between schematics and the font number used inside the pie charts is very small.

>> Typo corrected. Thanks.

We have also attempted an improved figure design to address the empty space and the small fonts.

2. Page 5, line 96 and 97. The authors do not acknowledge the existence of tools designed to detect genome-editing associated translocations (“in the context of this work” might need clarification). Moreover, the sentence syntax needs to be changed.

>> We thank the reviewer for pointing this out. We included a more detailed overview of the existing tools (introduction lines# 78-82, and in the discussion lines# 373-381) that support the identification of translocations associated with genome editing.

3. Page 7, line 161. The percentage indicated in the text do not match the percentage in the legend of Figure 2.

>> We thank the reviewer for noticing this. The numbers were corrected in the text of the results section- line# 180.

4. Page 12, line 257. Last sentence of the paragraph is missing a preposition.

>> The word “in” was added. Thanks.

5. Supplemental Figure 3 is not clear. Indel editing activity of what? Off-targets? Does n equal the number of events detected, the number of experiments performed on each off-target, or the number of off-targets? The legend does not indicate why red dots represent 0% (presumably those Off-targets with no discernable editing). Why does *RAG1* have two sites with 0 edit distances?

>> We thank the reviewer for highlighting that the legend description is lacking details. We included more clarification in the caption of Supplemental Figure 3. n represents the number of on- and off-target sites identified by the GUIDE-seq assay. Regarding the two sites of *RAG1* that have 0 edit distances, one is the on-target (*RAG1* locus), and one is an off-target with no mismatches. This off-target which perfectly matches the on-target does not have a PAM sequence. It is therefore not predicted to be a potential off-target site by standard tools like DESKGEN™ CRISPR.

6. With regard to the translocations detected overview in Supplemental Figure 7, does the hypergeometric test account for increased enrichment of intrachromosomal translocations due to increased interactions in cis?

>> CRISPECTOR is not taking into account the increased probability of intrachromosomal translocations in its detection. The HG test treats all possible translocations and structural variations detectable by the rhAmpSeq panel in the same way.

Further on the topic of translocations, are the authors taking into account the inability to sequence commonly primed (i.e. P5/P5 and P7/P7) illumina sequence tails and thus underestimating the translocation frequency by as much as 2-fold? This should be discussed given that the authors displayed the detectable combinations in Supplemental Figure 9 but failed to note this caveat in their frequency calculations.

>> We thank the reviewer for raising this very important point. In future CRISPECTOR versions, this shortcoming can be resolved by synthesizing the original PCR1 rhAmpSeq primers with a mix of altered 5' tail adapters (i.e P5&P7 on both reverse and fwd primers). This approach will guarantee detectability of all four configurations at every locus. We emphasized this point in the discussion (lines# 408-413) and in the results sections (lines# 276-281).

7. Supplemental Figure 9 has a typo in the legend.

>> Indeed. Corrected. Thanks.

8. Supplemental Table 2. The legend contains an error "Error! Reference source not found".

>> Reference error fixed.

9. In Supplementary Note S2, "Furthermore, this step was taken to make sure that reads were not overlooked due to the primer-matching algorithm that is described in Supplementary Note 6.". The reference should be to Supplementary Note S7.

>> Fixed.

10. Line 612: transcription  translocation.

>> Fixed.

11. Line 736: What is a WRS statistical analysis? It would appear based on the number of technical replicates presented that ddPCR really only confirmed the dominant translocation, which is in contrast to the statement in line 242 (experimentally confirmed which sites?); the statement is vague and does not represent the actual findings.

>> WRS - Wilcoxon Rank Sum is now referenced (Hollander and Wolfe, Wiley Interscience) - line# 812. The p-values are correct. It is true that only two of them fall below 0.05. Four of them fall below 0.1. This yields an FDR of 0.15.

Appendix1: Accounting for the effect of PCR duplication in Site 51 of RAG1, XT 2p in HEK-CAS9

Current approach – No PCR duplication filtering:

Final editing activity – 0.11%.

Conservative approach:

Given the assumption that any repetitive read in the mock experiment (except the amplicon reference read) is originated by a PCR duplication, we removed all identical reads from the mock. This approach can lead to FP estimation.

Final editing activity – 0.11%.

Although the final editing activity hasn't changed, note that in noisy positions, such as in the 1bp deletion at the cut-site, the mock indel frequency is now lower than the treatment indel frequency.

Anti-conservative approach:

Given the assumption that any repetitive read (except the amplicon reference read) is originated by a PCR duplication, we removed all identical reads from both the

treatment and the mock experiments.

Final editing activity – 0.061%.

With the PCR duplication filtering for both the treatment and the mock, the editing activity estimation decreased. The activity decreased due to the background (total read) estimation. Most reads belong to the amplicon reference sequence. There is no simple or obvious way to evaluate the PCR duplication for these reads. As a result, the number of edited indels, as called by CRISPECTOR, notably decreased, while the total number of reads only slightly decreased.

Reviewer #2 (Remarks to the Author):

In therapeutic gene editing, NGS amplicon sequencing is the gold standard for analysis of editing at off-target sites. Amit and Iancu et al. develop a software package, termed CRISPECTOR, for analyzing NGS amplicon sequencing experiments for CRISPR editing, focused on multiplex experiments for off-target analysis that involve comparisons between treated and control cells. They demonstrate improved accuracy examining editing on 5 pooled multiplex-PCR NGS experiments compared to CRISPResso2 and ampliCan. This is primarily validated by human expert examination and the dose-dependence of indels. Because multiple sets of primers are used in multiplexed PCR, one set for each site, the authors are able to look for mixing of primers between primer sets in resulting reads which indicate translocations. They use this translocation “module” to detect translocations in a number of experiments, which are partially validated by ddPCR.

The authors describe very clearly the limitations of existing packages and what they hope to accomplish with this new package. The key innovations compared to existing packages are 1) inference of edit percentages with confidence intervals at putative cut sites using a Bayesian inference approach and 2) a method for translocation detection at putative cut sites by examining reads. The improved inference of editing appears to be primarily important when editing percentages are very low or background noise is very high, so its utility is limited. The translocation detection method could be

helpful, although it requires a demonstration of sensitivity. Overall, the CRISPECTOR package may become a useful tool for the community, but a few points should be addressed with respect to the key claims:

>> We thank the reviewer for the accurate summary and appreciation of the value of this manuscript for the genome editing community.

Improved Inference of Editing:

1. Looking at the reads by hand, the examples highlighted in the paper all seem to be situations where even experts have to partially guess. I would have liked to see more than one expert validating these reads. Alternatively, blind expert validation where the expert is presented with unlabeled samples and told to identify whether editing occurred or not, or blinding where the expert is presented with pairs of reads, sometimes mock vs mock, would also suffice. The expert validation is not described in the methods.

>> The expert validation relies on the knowledge of which sample is the mock and which is the treatment. The assumption is that there is no editing in the mock sample but there are sequencing errors, and in the treatment samples, there are alignment errors. As the reviewer suggests, through the entire manual validation, two independent scientists examined the sequences and compared the indels to those located in the mock sample. We better clarified and elaborated in Supplemental Note S2- “Human validated dataset for the task of active off-target site classification”- lines # 489-605 and in the main text- lines# 325-326.

2. In most cases where CRISPResso2 seems to differ, including all statistically significant instances in Figure 2b, CRISPResso2 underestimates the editing percentage, primarily due to missed long indels. Missed long indels also account for mistakes with an alternative cut site. If the quantification window size parameter is varied for CRISPResso2, what are the results? If using a slightly larger window eliminates these discrepancies, then this is not a fair comparison. A varied parameter set is described for ampliCan but not for CRISPResso2 in Supplementary Note S6.

>> CRISPResso2 is the state of the art version of CRISPResso. In the CRISPResso2 follow-up article, the authors concluded that the recommended window around the predicted cleavage site is 2bps (1bp on each side). It may be the case that by bypassing the default settings results can change. In CRISPECTOR is not required to set these parameters (although they can). Instead – our more flexible approach produces the results automatically. Therefore, we see this as an important advantage of CRISPECTOR over CRISPResso2.

3. The text says “CRISPECTOR is designed to be highly sensitive, detecting events at rates as low as 0.1%. Such events occur quite often, as shown in Figure 1a and 1b.” To me, this implies that CRISPECTOR is detecting editing

rates as low as 0.1% accurately. However, Figure 1a and b instead show that, at certain cut sites, the indel rate for mock samples is sometimes higher than 0.1%. I would have expected to see a scatter plot of treatment vs. mock indel rate, a scatter plot of inferred editing rate vs. mock indel rate, or a histogram of inferred editing rate.

Indel frequencies for each type and position for site 51 of RAG1

>> We thank the reviewer for this important comment. We hereby added a scatter plot as requested, for off-target site 51 of *RAG1*, where each point represents read numbers (x axis for M and y axis for Tx) for an individual type and position combination. We are happy to add this as a Supplementary figure if the reviewer recommends that we do so.

4. In the 19 sites of disagreement between CRISPResso2 and CRISPECTOR, seeing dose dependence (or the lack of) at these sites would help solidify the edited/unedited claim.

>> We only have dose response experiments for some of the *EMX1* conditions. The calls of both CRISPECTOR and Crispresso2, at these sites and conditions are described in Supplemental Note 6. As we can see, for most sites we see concordance which holds for both tools. Three site/dose combinations constitute the 3 disagreements as reported in Table 1: (EMX1_7, crRNA, H), (EMX1_8, crRNA, H) and (EMX1_10, sg, H). While CRISPECTOR is concordant in all three of them, Crispresso2 is only concordant in two of them and is strictly discordant in EMX1_7, crRNA where L is called at 0.1 and H is called at 0.05. We do not have dose response data for other target/site combinations. We do hope, however, that the findings

reported in Supplemental Note S6 do, indeed, solidify the CRISPECTOR edit calls, albeit in small numbers. We also slightly modified the original text of Supplemental Note S6 to better clarify the message.

Minor comment:

1. There now exist a number of algorithms/tools for indel prediction (FORECasT, inDelphi, Lindel, ...). Is it possible to incorporate the results from these tools as priors for $p(\text{edit})$ for a specific indel outcome or as an aggregate for a particular nucleotide position, varying by gRNA?

>> Bioinformatics based priors can indeed be incorporated into the CRISPECTOR scheme. We will plan to do this in future versions. We do note that all tools known to us do not predict the indel type. Priors therefore have to be incorporated at later stages of the process.

2. The estimated background noise α is already quite conservative by choosing the 95th percentile of $\alpha(j)$, but in general it seems like it should vary from cut site to cut site, although I'm not sure how to do this rigorously. What does the ROC curve look like if you vary the percentile chosen for α ? It's also not clear to me what "The final $q(j)$ is averaged with a configurable pre-defined probability (0.9 in our experiments), which encourages stable estimation." means.

>> We do agree that the choice of working with the 95th percentile is somewhat arbitrary. A ROC curve is an excellent idea. In our case - however - the actual classification is not really known and we are using several indirect ways to validate results (human inspection, concordance etc). When more data is collected and when we may have better independent validation it will certainly make sense to re-examine parameters involved in computing $q(j)$. Regarding the final step in calculating $q(j)$ - our previous text was, indeed, too vague. We changed the text in the Supplementary Information- lines# 834-835. The idea of averaging with 0.9 is to stay on the conservative side.

3. <20/3000 reads is not something I would describe as a high rate in Supplementary Figure 4. Also, I assume 3 is supposed to be 13 in "18 of these 19 instances were determined, by expert investigation of the reads, to be CRISPResso2 errors – 5 false-positives and 3 false-negatives."

>> 29 out of 3000 is almost 1% which is high in the context of the off target framework. The typo was corrected, thanks for pointing it out.

Translocation Detection

1. The statistical analysis done here analyzing reads with the hypergeometric test is sound. However, to be a useful tool, the experimental side of this translocation module needs to have sufficient sensitivity to detect translocations adequately. To my knowledge, no one has shown that rhAmp-seq or similar methods can do this. A possible set of experiments to show this

include titrating small amounts of a synthesized “translocation product” into a multiplexed PCR containing templates for the standard amplicons.

>>We agree with the reviewer and value the reviewer’s suggestion. Following this recommendation, we performed a set of experiments measuring synthetic translocation constructs titrated into genomic DNA from edited cells. This design serves to evaluate the method’s sensitivity with respect to detecting translocations. The design of the translocation constructs was based on the CRISPECTOR calls, as obtained from the original experiment. We chose a pair of sites for which we did not detect a translocation. We included a description of the experiments and the conclusions in the methods (lines# 715-722), in the results (lines# 282-299) and in the discussion (lines# 321-324) sections. We also added a panel to Figure 3 (Panel d) that summarizes some of the results. The full results are presented in Supplemental Figure 12.

Some detected translocations are validated by ddPCR/regular PCR. I would have liked to see validation that translocations not detected by CRISPECTOR can not be found by ddPCR/regular PCR.

>>The reviewer raised an excellent point. We used ddPCR to measure the RAG2_10/RAG2_7 potential translocation (same which was used in the synthetic construct above) in edited *RAG2* HEK293-Cas9-XT crRNA cells. The ddPCR primers were designed to flank the ends of the potential fusion point. No translocation was detected in this experiment. A positive control, the 0.016% spike-in of the synthetic construct, was used to validate the ddPCR. We added text to the results section-lines# 294-296.

2. Along these lines, comparisons to existing methods would be good to include. LAM-HTGTS may be more sensitive due to the enzymatic blocking step eliminating amplicons without translocation, so UDiTaS may be the fairest comparison. UDiTaS can easily be augmented to include multiple primers in the first round of PCR, as mentioned in the discussion of the manuscript.

>> We agree with the reviewer that comparison of CRISPECTOR against other methods would improve the manuscript. A major advantage of the CRISPECTOR/rhAmpSeq approach is that it can identify translocations by using multiple anchors/off-target sites that had been previously identified. We agree that UDiTaS can potentially be augmented to include multiple primers in the first round of PCR. This would require extensive work, including optimization of the multiplex PCR, and at this point is beyond the scope of our paper. We also believe that the CRISPECTOR statistical approach can be adapted to analyze UDiTaS results. This can be the focus of a future project.

Minor comments:

3. The authors state that GUIDE-seq can not be used to detect structural variant events. The original GUIDE-seq paper shows translocation detection in their AMP-seq add-on in Figure 5, detecting all 4 types of translocations between an on-target site and an off-target site.

>> We thank the reviewer for this comment. Indeed, the original GUIDE-seq paper included an AMP-seq add-on method for translocation detection. We have corrected this in the revised manuscript- lines# 78-82.

4. When inferring editing activity, the package is able to detect alternative cut sites. Is this alternative cut site information used in generating the template for read alignment for translocations? The methods were not clear on this point.

>> CRISPECTOR detects reads that are putatively originating from translocations by matching read prefixes and suffixes to forward and reverse primers (Supplemental Figure 8) and therefore templates for read alignment are actually not being produced.

5. Can CRISPECTOR be quantitative beyond detection/existence of a particular translocation? It looks like there is a correlation between translocation read counts and translocation activity, but showing this definitively would strengthen the paper.

>> We thank the reviewer for this comment. Indeed, CRISPECTOR, in its current form does not attempt to quantify the translocation rates. However, having performed the titration experiments, following the reviewer's recommendations, we can demonstrate a positive correlation between the level of spiked-in translocation constructs/molecules and the number of reads detected by CRISPECTOR. This is reported in the newly added Figure 3d and in related text (lines# 297-298). We do not think, nonetheless, that this should be emphasized or turned into a feature at this point. The observed correlation needs to be much further developed in order to provide any quantitative results.

6. I think the heat-maps would show more useful information if they were displayed as a symmetric matrix, with everything under the diagonal being read counts for statistically significant translocations from mock samples, and everything above the diagonal being read counts from treatment samples.

>> We thank the reviewer for this suggestion. Based on internal consultation (including the team who created the report) we feel that this additional information may be confusing for the user at this higher level of the report generated. The M read numbers are provided in a table format but not in the heatmap. Note that we are only reporting translocation pairs with HG p-val < 0.05 in the heatmap. This makes the value of M somewhat redundant.

Other minor comments:

1. XT 2p is mentioned in multiple figures in the main text as well as the supplement but is not described anywhere in the text. What is XT 2p? crRNA XT:tracrRNA duplex?

>> We have described the crRNA XT:tracrRNA duplex gRNA in the revised manuscript under the section called gRNA synthesis, in the methods section-lines# 634-638.

2. The tool itself has a nice user interface and is easy to use. One suggestion for improvement is to integrate the edited reads table directly into the main output of the HTML report (so you don't have to click on it), and to format the table such that reads with insertions are shown properly aligned to the target site. Examples of other tools where this is done include inDelphi and OutKnocker.

>> We thank the reviewer for these suggestions related to the tool and the UI. Loading the read table does take a few seconds and to support smoother interactions, and since not all users are interested in the raw data of the read table, we have decided that a link mechanism is preferable. Regarding alignments of insertions - these are always problematic when we want to keep the consensus genomic sequence fixed. We chose the same approach, as did CRISPRESSO2. This may be revised in the future.

Reviewers' Comments:

Reviewer #1:

Remarks to the Author:

The limitations of the technique are stated more clearly, and the major/minor points are for the most part addressed. The addition of the assessment to Appendix 1 as suggested by the authors will be helpful for groups interested in using this approach. However, the field will certainly benefit from discerning PCR duplication at this resolution and urge the authors to continue improving down this path to incorporate an updated version in the near future. Overall, the authors' responses and changes to the manuscript significantly improved clarity and will help to engage the readership further.

Reviewer #2:

Remarks to the Author:

The revised manuscript is satisfactory.

Point-by-point response to the reviewer's comments

We would like to thank the reviewers for their careful and positive evaluation of our revised manuscript. We have included Supplemental note 11 as was requested by Reviewer #1. We also agree with Reviewer #1 that the field will benefit from discerning PCR duplication and we are planning to include support for UMIs to offset much of this gap in future versions of CRISPECTOR and in future protocols for measuring editing activity. Thank you again for your assistance during this process. We look forward to publishing our work in Nature Communication.

Sincerely,

Zohar Yakhini

Computer Science Department, Technion, Haifa, Israel

School of Computer Science, IDC Herzliya, Herzliya, Israel

Ayal Hendel

Life Science faculty, Bar Ilan University, Ramat-Gan, Israel